# PRUNING NEURAL NETWORKS AT INITIALIZATION: WHY ARE WE MISSING THE MARK?

**Jonathan Frankle**
MIT CSAIL

**Gintare Karolina Dziugaite**
Element AI

**Daniel M. Roy**
University of Toronto
Vector Institute

**Michael Carbin**
MIT CSAIL

## ABSTRACT

Recent work has explored the possibility of pruning neural networks at initialization. We assess proposals for doing so: SNIP (Lee et al., 2019), GraSP (Wang et al., 2020), SynFlow (Tanaka et al., 2020), and magnitude pruning. Although these methods surpass the trivial baseline of random pruning, we find that they remain below the accuracy of magnitude pruning after training. We show that, unlike magnitude pruning after training, randomly shuffling the weights these methods prune within each layer or sampling new initial values preserves or improves accuracy. As such, the per-weight pruning decisions made by these methods can be replaced by a per-layer choice of the fraction of weights to prune. This property suggests broader challenges with the underlying pruning heuristics, the desire to prune at initialization, or both.

## 1    INTRODUCTION

Since the 1980s, we have known that it is possible to eliminate a significant number of parameters from neural networks without affecting accuracy at inference-time (Reed, 1993; Han et al., 2015). Such neural network *pruning* can substantially reduce the computational demands of inference when conducted in a fashion amenable to hardware (Li et al., 2017) or combined with libraries (Elsen et al., 2020) and hardware designed to exploit sparsity (Cerebras, 2019; NVIDIA, 2020; Toon, 2020).

When the goal is to reduce inference costs, pruning often occurs late in training (Zhu & Gupta, 2018; Gale et al., 2019) or after training (LeCun et al., 1990; Han et al., 2015). However, as the financial, computational, and environmental demands of training (Strubell et al., 2019) have exploded, researchers have begun to investigate the possibility that networks can be pruned early in training or even before training. Doing so could reduce the cost of training existing models and make it possible to continue exploring the phenomena that emerge at larger scales (Brown et al., 2020).

There is reason to believe it may be possible to prune early in training without affecting final accuracy. Work on the *lottery ticket hypothesis* (Frankle & Carbin, 2019; Frankle et al., 2020a) shows that, from early in training (although often after initialization), there exist subnetworks that can train in isolation to full accuracy (Figure 1, red line). These subnetworks are as small as those found by inference-focused pruning methods after training (Appendix E; Renda et al., 2020), raising the prospect that it may be possible to maintain this level of sparsity for much or all of training. However, this work does not suggest a way to find these subnetworks without first training the full network.

The pruning literature offers a starting point for finding such subnetworks efficiently. Standard networks are often so overparameterized that pruning randomly has little effect on final accuracy at lower sparsities (green line). Moreover, many existing pruning methods prune during training (Zhu & Gupta, 2018; Gale et al., 2019), even if they were designed with inference in mind (orange line).

Recently, several methods have been proposed specifically for pruning at initialization. SNIP (Lee et al., 2019) aims to prune weights that are least salient for the loss. GraSP (Wang et al., 2020) aims to prune weights that most harm or least benefit gradient flow. SynFlow (Tanaka et al., 2020) aims to iteratively prune weights with the lowest "synaptic strengths" in a data-independent manner with the goal of avoiding *layer collapse* (where pruning concentrates on certain layers).

In this paper, we assess the efficacy of these pruning methods at initialization. How do SNIP, GraSP, and SynFlow perform relative to each other and naive baselines like random and magnitude pruning?

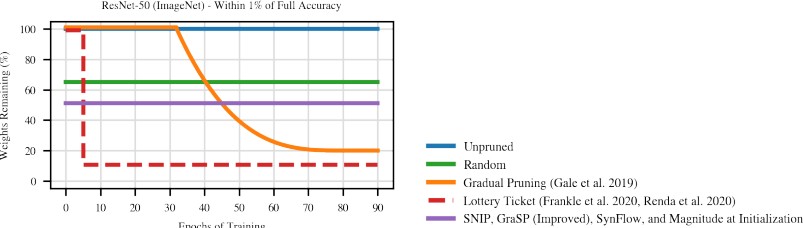

Figure 1: Weights remaining at each training step for methods that reach accuracy within one percentage point of ResNet-50 on ImageNet. Dashed line is a result that is achieved retroactively.

How does this performance compare to methods for pruning after training? Looking ahead, are there broader challenges particular to pruning at initialization? Our purpose is to clarify the state of the art, shed light on the strengths and weaknesses of existing methods, understand their behavior in practice, set baselines, and outline an agenda for the future. We focus at and near *matching sparsities*: those where magnitude pruning after training matches full accuracy.[1] We do so because: (1) these are the sparsities typically studied in the pruning literature, and (2) for magnitude pruning after training, this is a tradeoff-free regime where we do not have to balance the benefits of sparsity with sacrifices in accuracy. Our experiments (summarized in Figure 2) and findings are as follows:

**The state of the art for pruning at initialization.** The methods for pruning at initialization (SNIP, GraSP, SynFlow, and magnitude pruning) generally outperform random pruning. No single method is SOTA: there is a network, dataset, and sparsity where each pruning method (including magnitude pruning) reaches the highest accuracy. SNIP consistently performs well, magnitude pruning is surprisingly effective, and competition increases with improvements we make to GraSP and SynFlow.

**Magnitude pruning after training outperforms these methods.** Although this result is not necessarily surprising (after all, these methods have less readily available information upon which to prune), it raises the question of whether there may be broader limitations to the performance achievable when pruning at initialization. In the rest of the paper, we study this question, investigating how these methods differ behaviorally from standard results about pruning after training.

**Methods prune layers, not weights.** The subnetworks that these methods produce perform equally well (or better) when we randomly shuffle the weights they prune in each layer; it is therefore possible to describe a family of equally effective pruning techniques that randomly prune the network in these per-layer proportions. The subnetworks that these methods produce also perform equally well when we randomly reinitialize the unpruned weights. These behaviors are not shared by state-of-the-art weight-pruning methods that operate after training; both of these ablations (shuffling and reinitialization) lead to lower accuracy (Appendix F; Han et al., 2015; Frankle & Carbin, 2019).

**These results appear specific to pruning at initialization.** There are two possible reasons for the comparatively lower accuracy of these methods and for the fact that the resulting networks are insensitive to the ablations: (1) these behaviors are intrinsic to subnetworks produced by these methods or (2) these behaviors are specific to subnetworks produced by these methods at initialization. We eliminate possibility (1) by showing that using SNIP, SynFlow, and magnitude to prune the network after initialization leads to higher accuracy (Section 6) and sensitivity to the ablations (Appendix F). This result means that these methods encounter particular difficulties when pruning at initialization.

**Looking ahead.** These results raise the question of whether it is generally difficult to prune at initialization in a way that is sensitive to the shuffling or reinitialization ablations. If methods that maintain their accuracy under these ablations are inherently limited in their performance, then there may be broader limits on the accuracy attainable when pruning at initialization. Even work on lottery tickets, which has the benefit of seeing the network after training, reaches lower accuracy and is unaffected by these ablations when pruning occurs at initialization (Frankle et al., 2020a).

Although accuracy improves when using SNIP, SynFlow, and magnitude after initialization, it does not match that of the full network unless pruning occurs nearly halfway into training (if at all). This means that (1) it may be difficult to prune until much later in training or (2) we need new methods designed to prune early in training (since SNIP, GraSP, and SynFlow were not intended to do so).

---

[1]Tanaka et al. design SynFlow to avert *layer collapse*, which occurs at higher sparsities than we consider. However, they also evaluate at our sparsities, so we believe this is a reasonable setting to study SynFlow.

| Method | Early Pruning Methods | | | | | Baseline Methods | | | Ablations | | |
|---|---|---|---|---|---|---|---|---|---|---|---|
| | SNIP | GraSP | SynFlow | Magnitude | Random | LTR | Magnitude (After) | Other | Reinit | Shuffle | Invert |
| SNIP | — | — | — | ✗ | ✗ | ✗ | ✗ | ✗ | ✓ | ✗ | ✗ |
| GraSP | ✓ | — | — | ✗ | ✓ | ✓ | ✗ | ✓ | ✗ | ✗ | ✗ |
| SynFlow | ✓ | ✓ | — | ✓ | ✓ | ✗ | ✗ | ✗ | ✗ | ✗ | ✗ |

Figure 2: Comparisons in the SNIP, GraSP, and SynFlow papers. Does not include MNIST. SNIP lacks baselines beyond MNIST. GraSP includes random, LTR, and other methods; it lacks magnitude at init and ablations. SynFlow has other methods at init but lacks baselines or ablations.

## 2 RELATED WORK

Neural network pruning dates back to the 1980s (survey: Reed, 1993), although it has seen a recent resurgence (survey: Blalock et al., 2020). Until recently, pruning research focused on improving efficiency of inference. However, methods that *gradually prune* throughout training provide opportunities to improve the efficiency of training as well (Zhu & Gupta, 2018; Gale et al., 2019).

Lottery ticket work shows that there are subnetworks before (Frankle & Carbin, 2019) or early in training (Frankle et al., 2020a) that can reach full accuracy. Several recent papers have proposed efficient ways to find subnetworks at initialization that can train to high accuracy. SNIP (Lee et al., 2019), GraSP (Wang et al., 2020), SynFlow (Tanaka et al., 2020), and NTT (Liu & Zenke, 2020) prune at initialization. De Jorge et al. (2020) and Verdenius et al. (2020) apply SNIP iteratively; Cho et al. (2020) use SNIP for pruning for inference. Work on *dynamic sparsity* maintains a pruned network throughout training but regularly changes the sparsity pattern (Mocanu et al., 2018; Dettmers & Zettlemoyer, 2019; Evci et al., 2019); to match the accuracy of standard methods for pruning after training, Evci et al. needed to train for five times as long is standard for training the unpruned networks. You et al. (2020) prune after some training; this research is not directly comparable to any of the aforementioned papers as it prunes channels (rather than weights as in all work above) and does so later (20 epochs) than SNIP/GraSP/SynFlow (0) and lottery tickets (1-2).

## 3 METHODS

**Pruning.** Consider a network with weights $w_\ell \in \mathbb{R}^{d_\ell}$ in each layer $\ell \in \{1, \ldots, L\}$. Pruning produces binary *masks* $m_\ell \in \{0, 1\}^{d_\ell}$. A pruned *subnetwork* has weights $w_\ell \odot m_\ell$, where $\odot$ is the element-wise product. The *sparsity* $s \in [0, 1]$ of the subnetwork is the fraction of weights pruned: $1 - \sum_\ell m_\ell / \sum_\ell d_\ell$. We study pruning methods $\mathsf{prune}(W, s)$ that prune to sparsity $s$ using two operations. First, $\mathsf{score}(W)$ issues *scores* $z_\ell \in \mathbb{R}^{d_\ell}$ to all weights $W = (w_1, \ldots, w_L)$. Second, $\mathsf{remove}(Z, s)$ converts scores $Z = (z_1, \ldots, z_L)$ into masks $m_\ell$ with overall sparsity $s$. Pruning may occur in *one shot* (score once and prune from sparsity 0 to $s$) or *iteratively* (repeatedly score unpruned weights and prune from sparsity $s^{\frac{n-1}{N}}$ to $s^{\frac{n}{N}}$ over iterations $n \in \{1, \ldots, N\}$).

**Re-training after pruning.** After pruning at step $t$ of training, we subsequently train the network further by repeating the entire learning rate schedule from the start (Renda et al., 2020). Doing so ensures that, no matter the value of $t$, the pruned network will receive enough training to converge.

**Early pruning methods.** We study the following methods for pruning early in training.

*Random.* This method issues each weight a random score $z_\ell \sim \mathsf{Uniform}(0, 1)$ and removes weights with the lowest scores. Empirically, it prunes each layer to approximately sparsity $s$. Random pruning is a naive method for early pruning whose performance any new proposal should surpass.

*Magnitude.* This method issues each weight its magnitude $z_\ell = |w_\ell|$ as its score and removes those with the lowest scores. Magnitude pruning is a standard way to prune after training for inference (Janowsky, 1989; Han et al., 2015) and is an additional naive point of comparison for early pruning.

*SNIP (Lee et al., 2019).* This method samples training data, computes gradients $g_\ell$ for each layer, issues scores $z_\ell = |g_\ell \odot w_\ell|$, and removes weights with the lowest scores in one iteration. The justification for this method is that it preserves weights with the highest "effect on the loss (either positive or negative)." For full details, see Appendix B. In Appendix G, we consider a recently-proposed iterative variant of SNIP (de Jorge et al., 2020; Verdenius et al., 2020).

*GraSP (Wang et al., 2020).* This method samples training data, computes the Hessian-gradient product $h_\ell$ for each layer, issues scores $z_\ell = -w_\ell \odot h_\ell$, and removes weights with the highest scores

in one iteration. The justification for this method is that it removes weights that "reduce gradient flow" and preserves weights that "increase gradient flow." For full details, see Appendix C.

*SynFlow (Tanaka et al., 2020).* This method replaces the weights $w_\ell$ with $|w_\ell|$. It computes the sum $R$ of the logits on an input of 1's and the gradients $\frac{dR}{dw_\ell}$ of $R$. It issues scores $z_\ell = |\frac{dR}{dw_\ell} \odot w_\ell|$ and removes weights with the lowest scores. It prunes iteratively (100 iterations). The justification for this method is that it meets criteria that ensure (as proved by Tanaka et al.) it can reach the maximal sparsity before a layer must become disconnected. For full details, see Appendix D.

**Benchmark methods.** We use two benchmark methods to illustrate the highest known accuracy attainable when pruning to a particular sparsity in general (not just at initialization). Both methods reach similar accuracy and match full accuracy at the same sparsities (Appendix E). Note: we use one-shot pruning, so accuracy is lower than in work that uses iterative pruning. We do so to make a fair comparison to the early pruning methods, which do not get to train between iterations.

*Magnitude pruning after training.* This baseline applies magnitude pruning to the weights at the end of training. Magnitude pruning is a standard method for one-shot pruning after training (Renda et al., 2020). We compare the early pruning methods at initialization against this baseline.

*Lottery ticket rewinding (LTR).* This baseline uses the mask from magnitude pruning after training and the weights from step $t$. Frankle et al. (2020a) show that, for $t$ early in training and appropriate sparsities, these subnetworks reach full accuracy. This baseline emulates pruning at step $t$ with an oracle with information from after training. Accuracy improves for $t > 0$, saturating early in training (Figure 6, blue). We compare the early pruning methods after initialization against this baseline.

**Sparsities.** We divide sparsities into three ranges (Frankle et al., 2020a). *Trivial sparsities* are the lowest sparsities: those where the network is so overparameterized that randomly pruning at initialization can still reach full accuracy. *Matching sparsities* are moderate sparsities: those where the benchmark methods can match the accuracy of the unpruned network. *Extreme sparsities* are those beyond. We focus on matching sparsities and the lowest extreme sparsities. Trivial sparsities are addressed by random pruning. Extreme sparsities require making subjective or context-specific tradeoffs between potential efficiency improvements of sparsity and severe drops in accuracy.

**Networks, datasets, and replicates.** We study image classification. It is the main (SNIP) or sole (GraSP and SynFlow) task in the papers introducing the early pruning methods and in the papers introducing modern magnitude pruning (Han et al., 2015) and LTR (Frankle et al., 2020a). We use ResNet-20 and VGG-16 on CIFAR-10, ResNet-18 on TinyImageNet, and ResNet-50 on ImageNet. See Appendix A for hyperparameters. We repeat experiments five (CIFAR-10) or three (TinyImageNet and ImageNet) times with different seeds and plot the mean and standard deviation.

## 4 PRUNING AT INITIALIZATION

In this section, we evaluate the early pruning methods at initialization. Figure 3 shows the performance of magnitude pruning (green), SNIP (red), GraSP (purple), and SynFlow (brown) at initialization. For context, it also includes the accuracy of pruning after training (blue), random pruning at initialization (orange), and the unpruned network (gray).

**Matching sparsities.** For matching sparsities,[2] SNIP, SynFlow and magnitude dominate depending on the network. On ResNet-20, all methods are similar but magnitude reaches the highest accuracy. On VGG-16, SynFlow slightly outperforms SNIP until 91.4% sparsity, after which SNIP overtakes it; magnitude and GraSP are at most 0.4 and 0.9 percentage points below the best method. On ResNet-18, SNIP and SynFlow remain even until 79.0% sparsity, after which SNIP dominates; magnitude and GraSP are at most 2.1 and 2.6 percentage points below the best method. On ResNet-50, SNIP, SynFlow, and magnitude perform similarly, 0.5 to 1 percentage points above GraSP.

In some cases, methods are able to reach full accuracy at non-trivial sparsities. On VGG-16, SNIP and SynFlow do so until 59% sparsity (vs. 20% for random and 93.1% for magnitude after training). On ResNet-18, SNIP does so until 89.3% sparsity and SynFlow until 79% sparsity (vs. 20% for random and 96.5% for magnitude after training). On ResNet-20 and ResNet-50, no early pruning

---

[2]Matching sparsities are those where magnitude pruning after training matches full accuracy. These are sparsities $\leq$ 73.8%, 93.1%, 96.5%, and 67.2% for ResNet-20, VGG-16, ResNet-18, and ResNet-50.

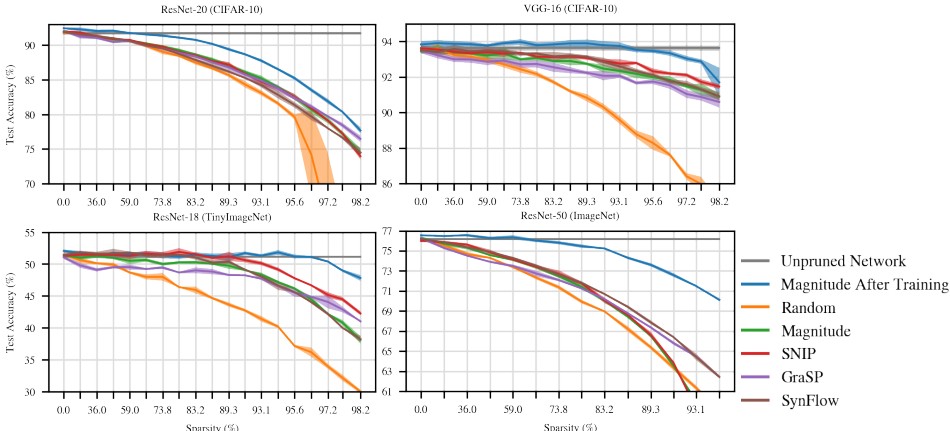

Figure 3: Accuracy of early pruning methods when pruning at initialization to various sparsities.

methods reach full accuracy at non-trivial sparsities; one possible explanation for this behavior is that these settings are more challenging and less overparameterized in the sense that magnitude pruning after training drops below full accuracy at lower sparsities.

**Extreme sparsities.** At extreme sparsities,[3] the ordering of methods is the same on VGG-16 but changes on the ResNets. On ResNet-20, magnitude and SNIP drop off, GraSP overtakes the other methods at 98.2% sparsity, and SynFlow performs worst; ResNet-50 shows similar behavior except that SynFlow performs best. On ResNet-18, GraSP overtakes magnitude and SynFlow but not SNIP.

**Summary.** No one method is SOTA in all settings and sparsities. SNIP consistently performs well, with SynFlow frequently competitive. Magnitude is surprisingly effective against more complicated heuristics. GraSP performs worst at matching sparsities but does better at extreme sparsities. Overall, the methods generally outperform random pruning; however, they cannot match magnitude pruning after training in terms of either accuracy or the sparsities at which they match full accuracy.

## 5    ABLATIONS AT INITIALIZATION

In this section, we evaluate the information that each method extracts about the network at initialization in the process of pruning. Our goal is to understand how these methods behave in practice and gain insight into why they perform differently than magnitude pruning after training.

**Randomly shuffling.** We first consider whether these pruning methods prune specific connections. To do so, we randomly shuffle the pruning mask $m_\ell$ within each layer. If accuracy is the same after shuffling, then the per-weight decisions made by the method can be replaced by the per-layer fraction of weights it pruned. If accuracy changes, then the method has determined which parts of the network to prune at a smaller granularity than layers, e.g., neurons or individual connections.

*Overall.* All methods maintain accuracy or improve when randomly shuffled (Figure 4, orange line). In other words, the useful information these techniques extract is not which individual weights to remove, but rather the layerwise proportions by which to prune the network.[4] Although layerwise proportions are an important hyperparameter for inference-focused pruning methods (He et al., 2018; Gale et al., 2019), proportions alone are not sufficient to explain the performance of those methods. For example, as the bottom row of Figure 4 shows, magnitude pruning after training makes pruning decisions specific to particular weights; shuffling in this manner reduces performance (Frankle & Carbin, 2019). In general, we know of no state-of-the-art weight-pruning methods (among the many that exist (Blalock et al., 2020)) with accuracy robust to this ablation. This raises the question of whether the insensitivity to shuffling for the early pruning methods may limit their performance.

---

[3]Extreme sparsities are those beyond which magnitude pruning after training reaches full accuracy.

[4]Note that there are certainly pathological cases where these proportions alone are not sufficient to match the accuracy of the pruning techniques. For example, at the extreme sparsities studied by Tanaka et al. (2020), pruning randomly could lead the network to become disconnected. An unlucky random draw could also lead the network to become disconnected. However, we do not observe these behaviors in any of our experiments.

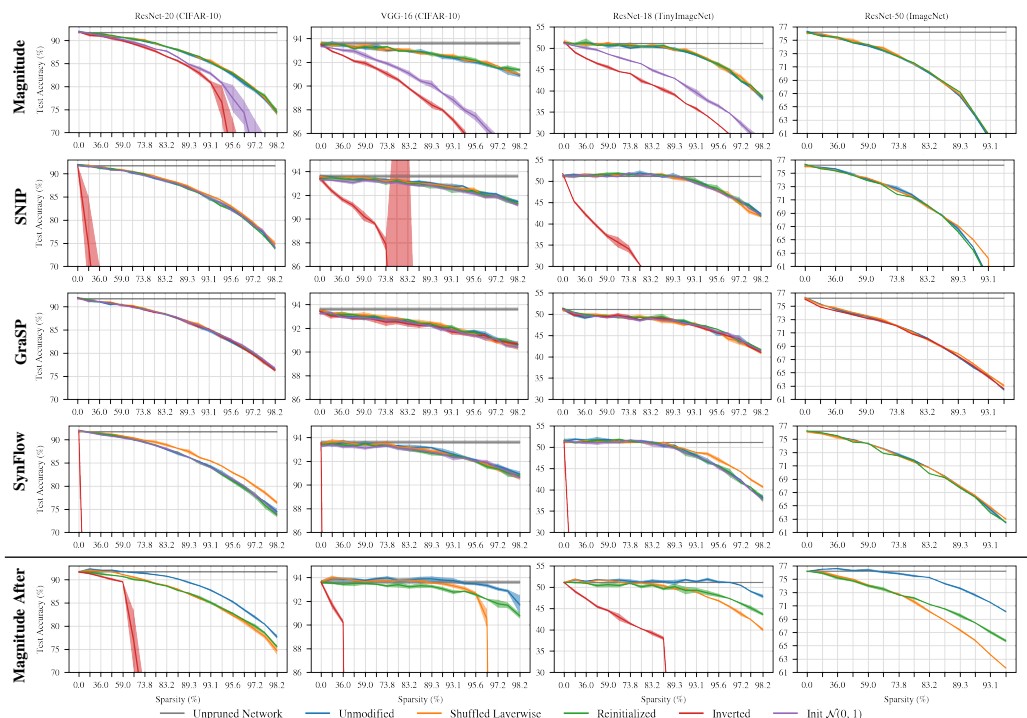

Figure 4: Ablations on subnetworks found by applying magnitude pruning, SNIP, GraSP, and Syn-Flow at initialization. (We ran limited ablations on ResNet-50 due to resource limitations.)

*Magnitude pruning.* Since the magnitude pruning masks can be shuffled within each layer, its pruning decisions are sensitive only to the per-layer initialization distributions. These distributions are chosen using He initialization: normal with a per-layer variance determined by the fan-in or fan-out (He et al., 2015). These variances alone, then, are sufficient information to attain the performance of magnitude pruning—performance often competitive with SNIP, GraSP, and SynFlow. Without this information, magnitude pruning performs worse (purple line): if each layer is initialized with variance 1, it will prune all layers by the same fraction no differently than random pruning. This does not affect SNIP,[5] GraSP, or SynFlow, showing a previously unknown benefit of these methods: they maintain accuracy in a case where the initialization is not informative for pruning in this way.

*SynFlow.* On ResNet-20 and ResNet-18, SynFlow accuracy improves at extreme sparsities when shuffling. We connect this behavior to a pathology of SynFlow that we term *neuron collapse*: SynFlow prunes entire neurons (in this case, convolutional channels) at a higher rate than other methods (Figure 5). At the highest matching sparsities, SynFlow prunes 31%, 52%, 69%, and 29% of neurons on ResNet-20, VGG-16, and ResNet-18, and ResNet-50. In contrast, SNIP prunes 5%, 11%, 32%, and 7%; GraSP prunes 1%, 6%, 14%, and 1%; and magnitude prunes 0%, 0%, < 1%, and 0%. Shuffling SynFlow layerwise reduces these numbers to 1%, 0%, 3.5%, and 13%[6] (orange line).

We believe neuron collapse is inherent to SynFlow. From another angle, SynFlow works as follows: consider all paths $p = \{w_p^{(\ell)}\}_\ell$ from any input node to any output node. The SynFlow gradient $\frac{dR}{dw}$ for weight $w$ is the sum of the products $\prod_\ell |w_p^{(\ell)}|$ of the magnitudes on all paths containing $w$. This is the weight's contribution to the network's ($\ell_{1,1}$) *path norm* (Neyshabur et al., 2015), a connection we demonstrate in Appendix D.5. Once an outgoing (or incoming) weight is pruned from a neuron, all incoming (or outgoing) weights are in fewer paths, decreasing $\frac{dR}{dw}$; they are more likely to be pruned on the next iteration, potentially creating a vicious cycle that prunes the entire neuron. Similarly, SynFlow heavily prunes skip connection weights, which participate in fewer paths (Appendix J).

**Reinitialization.** We next consider whether the networks produced by these methods are sensitive to the specific initial values of their weights. That is, is performance maintained when sampling a new

---

[5]Note that initialization can still affect the performance of SNIP. In particular, Lee et al. (2020) demonstrate a scheme for adjusting the initialization of a SNIP-pruned network after pruning that leads to higher accuracy.

[6]This number remains high for ResNet-50 because nearly half of the pruned neurons are in layers that get pruned entirely, specifically skip connections that downsample using 1x1 convolutions (see Appendix J).

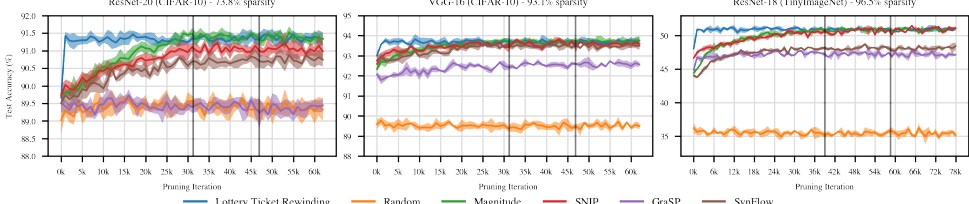

Figure 5: Percent of neurons (conv. channels) with sparsity $\geq s\%$ at the highest matching sparsity.

Figure 6: Accuracy of early pruning methods when pruning at the iteration on the x-axis. Sparsities are the highest matching sparsities. LTR prunes after training and initializes to the weights from the specified iteration. Vertical lines are iterations where the learning rate drops by 10x.

initialization for the pruned network from the same distribution as the original network? Magnitude pruning after training and LTR are known to be sensitive this ablation: when reinitialized, pruned networks train to lower accuracy (Figure 4 bottom row; Appendix F; Han et al., 2015; Frankle & Carbin, 2019), and we know of no state-of-the-art weight-pruning methods with accuracy robust to this ablation. However, all early pruning techniques are unaffected by reinitialization (green line): accuracy is the same whether the network is trained with the original initialization or a newly sampled initialization. As with random shuffling, this raises the question of whether this insensitivity to initialization may pose a limit to these methods that restricts performance.

**Inversion.** SNIP, GraSP, and SynFlow are each based on a hypothesis about properties of the network or training that allow a sparse network to reach high accuracy. Scoring functions should rank weights from most important to least important according to these hypotheses, making it possible to preserve the most important weights when pruning to any sparsity. In this ablation, we assess whether the scoring functions successfully do so: we prune the *most* important weights and retain the *least* important weights. If the hypotheses behind these methods are correct and they are accurately instantiated as scoring functions, then *inverting* in this manner should lower performance.

Magnitude, SNIP, and SynFlow behave as expected: when pruning the most important weights, accuracy decreases (red line). In contrast, GraSP's accuracy does not change when pruning the most important weights. This result calls into question the premise behind GraSP's heuristic: one can keep the weights that, according to Wang et al. (2020), *decrease* gradient flow the most and get the same accuracy as keeping those that purportedly increase it the most. Moreover, we find that pruning weights with the lowest-magnitude GraSP scores improves accuracy (Appendix H).

**Summary.** When using the methods for pruning at initialization, it is possible to reinitialize or layerwise shuffle the unpruned weights without hurting accuracy. This suggests that the useful part of the pruning decisions of SNIP, GraSP, SynFlow, and magnitude at initialization are the layerwise proportions rather than the specific weights or values. In the broader pruning literature, shuffling and reinitialization are associated with lower accuracy, and—as an example—we show that this is the case for magnitude pruning after training. Since these ablations do not hurt the accuracy of the methods for pruning at initialization, this raises the question of whether these methods may be confined to this lower stratum of performance.

This behavior under the ablations also raises questions about the reasons why SNIP, GraSP, and SynFlow are able to reach the accuracies in Section 4. Each paper poses a hypothesis about the qualities that would allow a pruned network to train effectively and derives a heuristic to determine which specific weights should be pruned based on this hypothesis. In light of our ablations, it is unclear whether the results achieved by these methods are attributable to these hypotheses. For example, SNIP aims to "identify important connections" (Lee et al., 2019); however, accuracy does not change if the pruned weights are randomly shuffled. SynFlow focuses on paths, "taking the inter-layer interactions of parameters into account" (Tanaka et al., 2020); accuracy improves in some

cases if we discard path information by shuffling and is maintained if we alter the "synaptic strengths flowing through each parameter" by reinitializing. In addition to these concerns, GraSP performs identically when inverted. Future early pruning research should use these and other ablations to evaluate whether the proposed heuristics behave according to the claimed justifications.

## 6  PRUNING AFTER INITIALIZATION

In Section 4, we showed that pruning at initialization leads to lower accuracy than magnitude pruning after training. In Section 5, we showed that this accuracy is invariant to ablations that hurt the accuracy of magnitude pruning after training. In this section, we seek to distinguish whether these behaviors are (1) intrinsic to the pruning methods or (2) specific to using the pruning methods at initialization. We have already shown evidence in support of (2) for magnitude pruning: the subnetworks it finds are less accurate and maintain their accuracy under shuffling and reinitialization at initialization but not when pruning after training (Figure 4 in Section 5). Moreover, LTR performs best when pruning early in training rather than at initialization (Frankle et al., 2020a), potentially pointing to broader difficulties specific to pruning at initialization.

To eliminate possibility (1), we need to demonstrate that there are circumstances where these methods reach higher accuracy than they do when pruning at initialization and where they are sensitive to the ablations. We do so by using these methods to prune later in training: we train for $k$ iterations, prune using each technique, and then train further for the entire learning rate schedule (Renda et al., 2020).[7] If accuracy improves when pruning at iteration $k > 0$ and the methods become sensitive to the ablations, then we know that the behaviors we observe in Sections 4 and 5 are not intrinsic to the methods. We note that SNIP, GraSP, and SynFlow were not designed to prune after initialization; as such, we focus on whether accuracy improves rather than the specific accuracy itself.

Figure 6 shows the effect of this experiment on accuracy. We include random pruning as a lower baseline and LTR is the best accuracy we know to be possible when applying a pruning mask early in training. Random pruning also provides a control to show that pruning after the network has trained for longer (and has reached higher accuracy) does not necessarily cause the pruned network to reach higher accuracy; no matter when in training we randomly prune, accuracy is the same.

Magnitude, SNIP, and SynFlow improve as training progresses, with magnitude and SNIP approaching LTR and SynFlow close behind. This means that the performance gap in Section 4 is not intrinsic to these methods, but rather is due to using them at initialization. Moreover, in Appendix F, we show that, as accuracy improves, the subnetworks are increasingly sensitive to the ablations, further evidence that these behaviors are specific to initialization.

Interestingly, LTR reaches higher accuracy than the pruning methods early in training: magnitude pruning does not match the accuracy of LTR at iterations 1K, 2K, and 1K until iterations 25K, 26K, and 36K on ResNet-20, VGG-16, and ResNet-18. If this gap in accuracy reflects a broader challenge inherent to pruning early in training, then these results suggest it may be difficult to prune, not just at initialization, but for a large period after.

## 7  DISCUSSION

**The state of the art.** We establish the following findings about pruning at initialization.

*Surpassing random pruning.* All methods surpass random pruning at some or all matching sparsities, and, in certain settings, some methods maintain full accuracy at non-trivial sparsities.

*No single method is SOTA at initialization.* Depending on the network, dataset, and sparsity, there is a setting where each early pruning method reaches the highest accuracy. Our enhancements to GraSP (Figure 13 in Appendix H) and SynFlow (Figure 4) further tighten the competition.

*Data is not currently essential at initialization.* SynFlow and magnitude pruning are competitive at initialization without using any training data. Like robustness to shuffling and reinitialization, however, data-independence may only be possible for the limited performance of current methods. In contrast, magnitude pruning after training and LTR rely on data for both pruning and initialization.

---

[7]Due to the expense of this experiment, we examine a single sparsity: the most extreme matching sparsity.

*Below the performance of pruning after training.* All methods for pruning at initialization reach lower accuracy than magnitude pruning after training.

**The challenge of pruning at initialization.** It is striking that methods that use such different signals (magnitudes; gradients; Hessian; data or lack thereof) reach similar accuracy, behave similarly under ablations, and improve similarly (except GraSP) when pruning after initialization.

*Ablations.* None of the methods we examine are sensitive to the specific weights that are pruned or their specific values when pruning at initialization. When pruning after training, these ablations are associated with lower accuracy. This contrast raises the question of whether methods that maintain accuracy under these ablations are inherently limited in the accuracy they can reach, and whether this is a property of these methods or of pruning at initialization in general. It may yet be possible to design a method that reaches higher accuracy and is sensitive to these ablations at initialization, but it may also be that the failures of four methods to do so (and their improvements after initialization in Section 6 and Appendix F) point to broader challenges in pruning at initialization.

*Why is it challenging?* We do not identify a cause for why these methods struggle to prune in a specific fashion at initialization (and often only at initialization), and we believe that this is an important question for future work. Perhaps there are properties of optimization that make pruning specific weights difficult or impossible at initialization (Evci et al., 2019). For example, training occurs in multiple phases (Gur-Ari et al., 2018; Jastrzebski et al., 2020; Frankle et al., 2020b); perhaps it is challenging to prune during this initial phase. Further study of this question may reveal whether this challenge is related to shared characteristics of these methods or whether it is intrinsic to pruning at initialization.

**Looking ahead.** We close by discussing the implications of our findings for future pruning research.

*Why prune early in training?* There are many reasons to do so. This includes the scientific goals of studying the capacity needed for learning (Frankle & Carbin, 2019) and the information needed to prune (Tanaka et al., 2020) and the practical goals of reducing the cost of finding pruned networks for inference (Lee et al., 2019) and "saving resources at training time" (Wang et al., 2020).

We focus on the practical goal of reducing the cost of training, since our results show that pruning at initialization is not a drop-in replacement for pruning after training. We find that existing methods for pruning at initialization require making a tradeoff: reducing the cost of training entails sacrificing some amount of accuracy. Looking ahead, we believe the most compelling next step for pruning research is developing tradeoff-free ways to reduce the cost of training.

*Pruning after initialization.* In Section 6, we show that SNIP, SynFlow, and magnitude improve gradually after initialization but LTR improves much faster. However, these methods were designed for initialization; focusing early in training may require new approaches. Alternatively, it may be that, even at iterations where LTR succeeds in Section 6, the readily available information is not sufficient reach this performance without consulting the state of the network after training. One way to avoid this challenge is to dynamically change the mask to exploit signals from later in training (Mocanu et al., 2018; Dettmers & Zettlemoyer, 2019; Evci et al., 2020).

*New signals for pruning.* It may be possible to prune at initialization or early in training, but signals like magnitudes and gradients (which suffice late in training) may not be effective. Are there different signals we should use early in training? Should we expect signals that work early in training to work late in training (or vice versa)? For example, second order information should behave differently at initialization and convergence, which may explain why GraSP struggles after initialization.

*Measuring progress.* We typically evaluate pruning methods by comparing their accuracies at certain sparsities. In the future, we will need to extend this framework to account for tradeoffs in different parts of the design space. At initialization, we must weigh the benefits of extreme sparsities against decreases in accuracy. This is especially important for methods like SynFlow and FORCE (de Jorge et al., 2020), which are designed to maintain diminished but non-random accuracy at the most extreme sparsities. In this paper, we defer this tradeoff by focusing on matching sparsities.

When pruning after initialization, we will need to address an additional challenge: comparing a method that prunes to sparsity $s$ at step $t$ against a method that prunes to sparsity $s' > s$ at step $t' > t$. To do so, we will need to measure overall training cost. That might include measuring the area under the curve in Figure 1, FLOPs (as, e.g., Evci et al. (2019) do), or real-world training time and energy consumption on software and hardware optimized for pruned neural networks.

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

## CONTENTS OF THE APPENDICES

**Appendix A.** The details of the networks, datasets, and hyperparameters we use in our experiments.

**Appendix B.** Details of our replication of SNIP.

**Appendix C.** Details of our replication of GraSP.

**Appendix D.** Details of our replication of SynFlow.

**Appendix E.** Plots of the baselines: random pruning, magnitude pruning at initialization, magnitude pruning after training, and LTR.

**Appendix F.** The shuffling and reinitialization ablations from Section 5 conducted when the pruning methods are applied *after* training.

**Appendix G.** The result of performing SNIP iteratively over 100 iterations rather than in one shot.

**Appendix H.** The three different variants of GraSP we study: pruning the weights with the lowest scores, highest scores, and lowest-magnitude scores.

**Appendix I.** A comparison of the early pruning methods including our improvements to GraSP and SynFlow.

**Appendix J.** The layerwise proportions in which each methods prunes the networks.

**Appendix K.** Experiments from the main body and the appendices on the LeNet-300-100 fully-connected network for MNIST.

**Appendix L.** Experiments from the main body and the appendices on a modified version of ResNet-18 based on the network used by Tanaka et al. (2020) in the SynFlow paper.

**Appendix ??.** The ICLR 2021 concurrent work policy.

**Appendix M.** An analysis of the *effective sparsities* of the networks when taking into account weights that are disconnected but that have not explicitly been pruned.

# A    Networks, Datasets, and Training

We use the following combinations of networks and datasets for image classification:

| Network | Dataset | Appears | Notes |
|---|---|---|---|
| LeNet-300-100 | MNIST | Appendix K | To evaluate the random shuffling ablation on a fully-connected network. |
| ResNet-20 | CIFAR-10 | Main Body | |
| VGG-16 | CIFAR-10 | Main Body | |
| ResNet-18 | TinyImageNet | Main Body | |
| Modified ResNet-18 | Modified TinyImageNet | Appendix L | To match the ResNet-18/TinyImageNet experiment in the SynFlow paper. |
| ResNet-50 | ImageNet | Main Body | |

## A.1    Networks

The networks are designed as follows:

- LeNet-300-100 is a fully-connected network with two hidden layers for MNIST. The first hidden layer has 300 units and the second hidden layer has 100 units. The network has ReLU activations.

- ResNet-20 is the CIFAR-10 version of ResNet with 20 layers as designed by He et al. (2016). We place batch normalization prior to activations. We use the ResNet-20 implementation from the OpenLTH repository.[8]

- VGG-16 is a CIFAR-10 network as described by Lee et al. (2019). The first two layers have 64 channels followed by 2x2 max pooling; the next two layers have 128 channels followed by 2x2 max pooling; the next three layers have 256 channels followed by 2x2 max pooling; the next three layers have 512 channels followed by max pooling; the final three layers have 512 channels. Each channel uses 3x3 convolutional filters. VGG-16 has batch normalization before each ReLU activation. We use the VGG-16 implementation from the OpenLTH repository.

- ResNet-18 and ResNet-50 are the ImageNet version of ResNet with 18 and 50 layers as designed by He et al. (2016). We use the TorchVision implementations of these networks.

- Modified ResNet-18 is a modified version of ResNet-18 designed to match the version used by Tanaka et al. (2020) in the SynFlow paper.[9] This version has been modified specifically for TinyImageNet: the first convolution has filter size 3x3 (rather than 7x7) and the max-pooling layer that follows has been eliminated. For more discussion, see Appendix D.

## A.2    Datasets

- CIFAR-10 is augmented by normalizing per-channel, randomly flipping horizontally, and randomly shifting by up to four pixels in any direction.

- TinyImageNet is augmented by normalizing per channel, selecting a patch with a random aspect ratio between 0.8 and 1.25 and a random scale between 0.1 and 1, cropping to 64x64, and randomly flipping horizontally.

- Modified TinyImageNet is augmented by normalizing per channel, randomly flipping horizontally, and randomly shifting by up to four pixels in any direction.

- ImageNet is augmented by normalizing per channel, selecting a patch with a random aspect ratio between 0.8 and 1.25 and a random scale between 0.1 and 1, cropping to 224x224, and randomly flipping horizontally.

## A.3    Training

| Network | Dataset | Epochs | Batch | Opt. | Mom. | LR | LR Drop | Weight Decay | Initialization | Iters per Ep | Rewind Iter |
|---|---|---|---|---|---|---|---|---|---|---|---|
| LeNet | MNIST | 40 | 128 | SGD | — | 0.1 | — | — | Kaiming Normal | 469 | 0 |
| ResNet-20 | CIFAR-10 | 160 | 128 | SGD | 0.9 | 0.1 | 10x at epochs 80, 120 | 1e-4 | Kaiming Normal | 391 | 1000 |
| VGG-16 | CIFAR-10 | 160 | 128 | SGD | 0.9 | 0.1 | 10x at epochs 80, 120 | 1e-4 | Kaiming Normal | 391 | 2000 |
| ResNet-18 | TinyImageNet | 200 | 256 | SGD | 0.9 | 0.2 | 10x at epochs 100, 150 | 1e-4 | Kaiming Normal | 391 | 1000 |
| Modified ResNet-18 | Modified TinyImageNet | 200 | 256 | SGD | 0.9 | 0.2 | 10x at epochs 100, 150 | 1e-4 | Kaiming Normal | 391 | 1000 |
| ResNet-50 | ImageNet | 90 | 1024 | SGD | 0.9 | 0.4 | 10x at epochs 30, 60, 80 | 1e-4 | Kaiming Normal | 1251 | 6255 |

---

[8] https://github.com/facebookresearch/open_lth
[9] https://github.com/ganguli-lab/Synaptic-Flow

# B    REPLICATING SNIP

In this Appendix, we describe and evaluate our replication of SNIP (Lee et al., 2019).

## B.1    ALGORITHM

SNIP introduces a virtual parameter $c_i \in 0, 1$ as a coefficient for each parameter $w_i$. Initially, SNIP assumes that $c_i = 1$.

SNIP assigns each parameter $w_i$ a score $s_i = \left| \frac{\partial L}{\partial c_i} \right|$ and prunes the parameters with the lowest scores.

This algorithm entails three key design choices:

1. Using the derivative of the loss with respect to $c_i$ as a basis for scoring.
2. Taking the absolute value of this derivative.
3. Pruning weights with the lowest scores.

Lee et al. (2019) explain these choices as follows: "if the magnitude of the derivative is high (regardless of the sign), it essentially means that the connection $c_i$ has a considerable effect on the loss (either positive or negative) and it has to be preserved to allow learning on $w_i$."

## B.2    IMPLEMENTATION DETAILS

**Algorithm.** We can rewrite the score as follows. Let $a_i$ be the incoming activation that is multiplied by $w_i$. Let $z$ be the pre-activation of the neuron to which $w_i$ serves as an input.

$$s_i = \left| \frac{\partial L}{\partial c_i} \right| = \left| \frac{\partial L}{\partial z} \frac{\partial z}{\partial c_i} \right| = \left| \frac{\partial L}{\partial z} a_i w_i \right| = \left| \frac{\partial L}{\partial z} \frac{\partial z}{\partial w_i} w_i \right| = \left| \frac{\partial L}{\partial w_i} w_i \right|$$

In summary, we can rewrite $s_i$ as the gradient of $w_i$ multiplied by the value of $w_i$. This is the formula we use in our implementation.

**Selecting examples for SNIP.** Lee et al. (2019) use a single mini-batch for SNIP. To create the mini-batch that we use for SNIP, we follow the strategy used by Wang et al. (2020) for GraSP: create a mini-batch composed of ten examples selected randomly from each class.

**Reinitializing.** After pruning, Lee et al. (2019) reinitialize the network.[10] We do not reinitialize.

**Running on CPU.** To avoid any risk that distributed training might affect results, we run all SNIP computation on CPU and subsequently train the pruned network on TPU.

---

[10]See discussion on OpenReview.

## B.3 NETWORKS AND DATASETS

Lee et al. consider the following networks for computer vision:

| SNIP Name | Our Name | Dataset | GitHub | Replicated | Notes |
|---|---|---|---|---|---|
| LeNet-300-100 | LeNet-300-100 | MNIST | ✓ | ✗ | Fully-connected |
| LeNet-5 | — | MNIST | ✓ | ✗ | Convolutional |
| AlexNet-s | — | CIFAR-10 | ✓ | ✗ | |
| AlexNet-b | — | CIFAR-10 | ✓ | ✗ | |
| VGG-C | — | CIFAR-10 | ✓ | ✗ | VGG-D but some layers have 1x1 convolutions |
| VGG-D | VGG2-16 | CIFAR-10 | ✓ | ✓ | VGG-like but with two fully-connected layers |
| VGG-like | VGG-16 | CIFAR-10 | ✓ | ✓ | VGG-D but with one fully-connected layer |
| WRN-16-8 | WRN-14-8 | CIFAR-10 | ✗ | ✓ | |
| WRN-16-10 | WRN-14-10 | CIFAR-10 | ✗ | ✓ | |
| WRN-22-8 | WRN-20-8 | CIFAR-10 | ✗ | ✓ | |

Table 1: The networks and datasets examined in the SNIP paper (Lee et al., 2019).

## B.4 RESULTS

A GitHub repository associated with the paper[11] includes all of those networks except for the wide ResNets (WRNs). We have made our best effort to replicate a subset of these networks in our research framework. Table 1 shows the results from our replication. Each of our numbers is the average across five replicates with different random seeds.

| Name | Unpruned Accuracy | | Sparsity | Pruned Accuracy | |
|---|---|---|---|---|---|
| | Reported | Ours | | Reported | Ours |
| VGG-16 | 91.7% | 93.6% | 97% | 92.0% $(+0.3)$ | 92.1% $(-1.5)$ |
| VGG2-16 | 93.2% | 93.5% | 95% | 92.9% $(-0.3)$ | 92.3% $(-1.2)$ |
| WRN-14-8 | 93.8% | 95.2% | 95% | 93.4% $(-0.4)$ | 93.4% $(-1.8)$ |
| WRN-14-10 | 94.1% | 95.4% | 95% | 93.6% $(-0.5)$ | 93.9% $(-1.4)$ |
| WRN-20-8 | 93.9% | 95.6% | 95% | 94.1% $(+0.3)$ | 94.3% $(-1.2)$ |

Table 2: The performance of SNIP as reported in the original paper and in our reimplementation.

**Unpruned networks.** The unpruned networks implemented by Lee et al. (2019) appear poorly tuned such that they do not achieve standard performance levels. The accuracy of our VGG-16 is 1.9 percentage points higher, and the accuracies of our wide ResNets are between 1.3 and 1.7 percentage points higher. In general, implementation details of VGG-style networks for CIFAR-10 vary widely (Blalock et al., 2020), so some differences are to be expected. However, ResNets for CIFAR-10 are standardized (He et al., 2016; Zagoruyko & Komodakis, 2016), and our accuracies are identical to those reported by Zagoruyko & Komodakis in the paper that introduced wide ResNets.

**Pruned networks.** After applying SNIP, our accuracies more closely match those reported in the paper. However, since our networks started at higher accuracies, these values represent much larger drops in performance than reported in the original paper. Overall, our results after applying SNIP appear to match those reported in the paper, giving us some confidence that our implementation is correct. However, since the accuracies of the unpruned networks in SNIP are lower than standard values, it is difficult to say for sure.

## B.5 RESULTS FROM GRASP PAPER

The paper that introduces GraSP (Wang et al., 2020) also replicates SNIP.[12] In Table 3, we compare their reported results with ours on the networks described in Table 4 in Appendix C.

---

[11]https://github.com/namhoonlee/snip-public/

[12]Although Wang et al. (2020) have released an open-source implementation of GraSP, this code does not include their implementation of SNIP. We are not certain which hyperparameters they used for SNIP.

| Name | Unpruned Accuracy | | | Results | |
| | Reported | Ours | Sparsity | Reported | Ours |
| --- | --- | --- | --- | --- | --- |
| VGG-19 | 94.2% | 93.5% | 90% | 93.6% (-0.6) | 93.5% (-0.0) |
| | | | 95% | 93.4% (-0.8) | 93.4% (-0.1) |
| | | | 98% | 92.1% (-2.1) | diverged |
| WRN-32-2 | 94.8% | 94.5% | 90% | 92.6% (-2.2) | 92.5% (-2.0) |
| | | | 95% | 91.1% (-3.7) | 91.0% (-3.5) |
| | | | 98% | 87.5% (-7.3) | 87.7% (-6.8) |
| ResNet-50 | 75.7% | 76.2% | 60% | 74.0% (-1.7) | 73.9% (-2.3) |
| | | | 80% | 69.7% (-6.0) | 71.2% (-5.0) |
| | | | 90% | 62.0% (-13.7) | 65.7% (-10.5) |

Table 3: The performance of SNIP as reported in the original paper and in our reimplementation.

**Unpruned networks.** See Appendix C.4 for a full discussion of the unpruned networks. Our VGG-19, WRN-32-2, and ResNet-50 reach slightly different accuracy than those of Wang et al. (2020), but the performance differences are much smaller than for the unpruned networks in Table 2.

**Pruned networks.** Although performance of our unpruned VGG-19 network is lower than that of Wang et al., the SNIP performances are identical at 90% and 95% sparsity. This outcome is similar to our results in Appendix B.4, where we found similar pruned performances despite the fact that unpruned performances differed. At 98% sparsity on VGG-19, three of our five runs diverged.

On ResNet-50 for ImageNet, our unpruned and SNIP accuracies are higher than those reported in by (Wang et al., 2020). This may be a result of different hyperparameter choices for training the network; see Appendix C.3 for full details. This may also be a result of different hyperparameter choices for SNIP. We may use a different number of examples than Wang et al. to compute the SNIP gradients and we may select these examples differently (although, since Wang et al. did not release their SNIP code, we cannot be certain); see Appendix C.2 for full details.

# C    REPLICATING GRASP

In this Appendix, we describe and evaluate our replication of GraSP (Wang et al., 2020).

## C.1    ALGORITHM

**Scoring parameters.** GraSP is designed to preserve the gradient flow through the sparse network that results from pruning. To do so, it attempts to prune weights in order to maximize the change in loss that takes place after the first step of training. Concretely, let $\Delta L(w)$ be the change in loss due to the first step of training:[13]

$$\Delta L(w) = L(w + \eta \cdot \nabla L(w)) - L(w)$$

where $\eta$ is the learning rate. Since GraSP focuses on gradient flow, it takes the limit as $\eta$ goes to 0:

$$\Delta L(w) = \lim_{\eta \to 0} \frac{L(w + \eta \cdot \nabla L(w)) - L(w)}{\eta} \approx \nabla L(w)^\top \nabla L(w) \tag{1}$$

The last expression emerge by taking a first-order Taylor expansion of $L(w + \eta \cdot \nabla L(w))$.

GraSP treats pruning the network as a perturbation $\delta$ transforming the original parameters $w$ into perturbed parameters $w + \delta$. The effect of this perturbation on the change in loss of the network can be measured by comparing $\Delta L(w + \delta)$ and $\Delta L(w)$:

$$C(\delta) = \Delta L(w + \delta) - \Delta L(w) = \nabla L(w + \delta)^\top \nabla L(w + \delta) - \nabla L(w)^\top \nabla L(w) \tag{2}$$

Finally, GraSP takes the first-order Taylor approximation of the left term about $w$, yielding:

$$
\begin{aligned}
C(\delta) &\approx \nabla L(w)^\top \nabla L(w) + 2\delta^\top \nabla^2 L(w) \nabla L(w) + O(||\delta||_2^2) - \nabla L(w)^\top \nabla L(w) \\
&= 2\delta^\top \nabla^2 L(w) \nabla L(w) + O(||\delta||_2^2) \\
&= 2\delta^\top H g
\end{aligned}
\tag{3}
$$

where $H$ is the Hessian and $g$ is the gradient. Pruning an individual parameter $w_i$ at index $i$ involves creating a vector $\delta^{(i)}$ where $\delta_i^{(i)} = -w_i$ and $\delta_j^{(i)} = 0$ for $j \neq i$. The resulting pruned parameter vector is $w - \delta^{(i)}$; this vector is identical to $w$ except that $w_i$ has been set to 0. Using the analysis above, the effect of pruning parameter $w_i$ in this manner on the gradient flow is approximated by $C(-\delta^{(i)}) = -w_i(Hg)_i$. GraSP therefore gives each weight the following score:

$$s_i = C(-\delta^{(i)}) = -w_i(Hg)_i \tag{4}$$

**Using scores to prune.** To use $s_i$ for pruning, Wang et al. make the following interpretation:

> *GraSP uses [Equation 3] as the measure of the importance of each weight. Specifically, if $C(\delta)$ is negative, then removing the corresponding weights will reduce gradient flow, while if it is positive, it will increase gradient flow.*

In other words, parameters with lower scores are more important (since removing them will have a less beneficial or more detrimental impact on gradient flow) and parameters with higher scores are less important (since removing them will have a more beneficial or less detrimental impact on gradient flow). Since the goal of GraSP is to maximize gradient flow after pruning, it should prune "those weights whose removal will not reduce the gradient flow," i.e., those with the highest scores.

---

[13] We believe this quantity should instead be specified as $L(w) - L(w - \eta \cdot \nabla L(w))$. The gradient update goes in the negative direction, so we should subtract the expression $\eta \cdot \nabla L(w)$ from the original initialization $w$. We expect loss to decrease after taking this step, so—if we want $\Delta L(w)$ to capture the improvement in loss—we need to subtract the updated loss from the original loss.

## C.2 IMPLEMENTATION DETAILS

**Algorithm.** To implement GraSP, we follow the PyTorch implementation provided by the authors on GitHub[14] (which computes the Hessian-gradient product according to Algorithm 2 of the paper).

**Selecting examples for scoring parameters.** To create the mini-batch that we use to compute the GraSP scores, we follow the strategy used by Wang et al. (2020) for the CIFAR-10 networks in their implementation: we randomly sample ten examples from each class. We use this approach for both CIFAR-10 and ImageNet; on ImageNet, this means we use 10,000 examples representing all ImageNet classes.

It is not entirely clear how Wang et al. select the mini-batch for the ImageNet networks in their experiments. In their configuration files, Wang et al. appear to use one example per class (1000 in total covering all classes). In their ImageNet implementation (which ignores their configuration files), they use 150 mini-batches where the batch size is 128 (19,200 examples covering an uncertain number of classes).

**Reinitializing.** We do not reinitialize after pruning.

**Running on CPU.** To avoid any risk that distributed training might affect results, we run all GraSP computation on CPU and subsequently train the pruned network on TPU.

## C.3 NETWORKS AND DATASETS

Wang et al. consider the networks for computer vision in Table 4 below. They use both CIFAR-10 and CIFAR-100 for all CIFAR-10 networks, while we only use CIFAR-10. Note that our hyperparameters for ResNet-50 on ImageNet differ from those in the GraSP implementation (likely due to different hardware): we use a larger batch size (1024 vs. 128), a higher learning rate (0.4 vs. 0.1).

| GraSP Name | Our Name | Dataset | GitHub | Replicated | Notes |
|---|---|---|---|---|---|
| VGG-19 | VGG-19 | CIFAR-10 | ✓ | ✓ | |
| ResNet-32 | WRN-32-2 | CIFAR-10 | ✓ | ✓ | Wang et al. use twice the standard width. |
| ResNet-50 | ResNet-50 | ImageNet | ✓ | ✓ | ResNets for CIFAR-10 and ImageNet are different. |
| VGG-16 | — | ImageNet | ✗ | ✗ | VGGs for CIFAR-10 and ImageNet are different. |

Table 4: The networks and datasets examined in the GraSP paper (Wang et al., 2020).

## C.4 RESULTS

The GitHub implementation of GraSP by Wang et al. (2020) includes all of the networks from Table 4 except VGG-16. We have made our best effort to replicate these networks in our research framework. Table C.2 shows the results from our replication. Each of our numbers is the average across five replicates with different random seeds.

**Unpruned networks.** Our unpruned networks perform similarly to those of Wang et al. (2020). Although we made every effort to replicate the architecture and hyperparameters of the GraSP implementation of VGG-19, our average accuracy is 0.7 percentage points lower.[15] Accuracy on WRN-32-2 is closer, differing by only 0.3 percentage points. Accuracy on ResNet-50 for ImageNet is higher by half a percentage point, likely due to the fact that we use different hyperparameters.

**Pruned networks.** Our pruned VGG-19 and WRN-32-2 also reach lower accuracy than those of Wang et al.; this difference is commensurate with the difference between the unpruned networks. On VGG-19, the accuracies of our pruned networks are lower than those of Wang et al. by 0.5 to 0.6 percentage points, matching the drop of 0.7 percentage points for the unpruned network. Similarly, on WRN-32-2, the accuracies of our pruned networks are lower than those of Wang et al. by 0.2 to 0.5 percentage points, matching the drop of 0.3 percentage points for the unpruned networks. In both cases, the decrease in performance after pruning (inside the parentheses in Table 5) is nearly identical between the two papers, differing by no more than 0.2 percentage points at any sparsity.

---

[14]https://github.com/alecwangcq/GraSP

[15]VGG networks for CIFAR-10 are notoriously difficult to replicate (Blalock et al., 2020).

| Name | Unpruned Accuracy | | Sparsity | Pruned Accuracy | |
|---|---|---|---|---|---|
| | Reported | Ours | | Reported | Ours |
| VGG-19 | 94.2% | 93.5% | 90%
95%
98% | 93.3% (−0.9)
93.0% (−1.2)
92.2% (−2.0) | 92.8% (−0.7)
92.5% (−1.0)
91.6% (−1.9) |
| WRN-32-2 | 94.8% | 94.5% | 90%
95%
98% | 92.4% (−2.4)
91.4% (−3.4)
88.8% (−6.0) | 92.2% (−2.3)
90.9% (−3.6)
88.3% (−6.2) |
| ResNet-50 | 75.7% | 76.2% | 60%
80%
90% | 74.0% (−1.7)
72.0% (−6.0)
68.1% (−7.6) | 73.4% (−2.8)
71.0% (−5.2)
67.0% (−9.2) |

Table 5: The performance of GraSP as reported in the original paper and in our reimplementation.

We conclude that the behavior of our implementation matches that of Wang et al., although we are starting from slightly lower baseline accuracy.

The accuracy of our pruned ResNet-50 networks is less consistent with that of Wang et al.. Despite starting from a higher baseline, our accuracy after pruning is lower by 0.6 to 1.1 percentage points. These differences are potentially due to different hyperparameters: as mentioned previously, we select examples for GraSP differently than Wang et al., and we train with a different batch size and learning rate.

## D  REPLICATING SYNFLOW

In this Appendix, we describe and evaluate our replication of SynFlow (Tanaka et al., 2020).

### D.1  ALGORITHM

SynFlow is an iterative pruning algorithm. It prunes to sparsity $s$ over the course of $N$ iterations, pruning from sparsity $s^{\frac{n-1}{N}}$ to sparsity $s^{\frac{n}{N}}$ on each iteration $n \in \{1, ..., N\}$. On each iteration, it issues scores to the remaining, unpruned weights and then removes those with the lowest scores.

Synflow scores weights as follows:

1. It replaces all parameters $w_\ell$ with their absolute values $|w_\ell|$.
2. It forward propagates an input of all ones through the network.
3. It computes the sum of the logits $R$.
4. It computes the gradient of $R$ with respect to each weight $|w|$: $\frac{dR}{dw}$.
5. It issues the score $|\frac{dR}{dw} \cdot w|$ for each weight.

Tanaka et al. (2020) explain these choices as follows. Their goal is to create a pruning technique that "provably reaches Maximal Critical Compression," i.e., a pruning technique that ensures that the network remains connected until the most extreme sparsity where it is possible to do so. As they prove, any pruning technique that is iterative, issues positive scores, and is *conservative* (i.e., the sum of the incoming and outgoing scores for a layer are the same), then it will reach maximum critical compression (Theorem 3). The iterative requirement is that scores are recalculated after each parameter is pruned; in their experiments, Tanaka et al. use 100 iterations in order to make the process more efficient.

### D.2  IMPLEMENTATION DETAILS

**Iterative pruning.** We use 100 iterations, the same as Tanaka et al. (2020) use (as noted in the appendices).

**Input size.** We use an input size that is the same as the input size for the corresponding dataset. For example, for CIFAR-10, we use an input that is 32x32x3; for ImageNet, we use an input that is 224x224x3.

**Reinitializing.** After pruning, Tanaka et al. (2020) do not reinitialize the network.

**Running on CPU.** To avoid any risk that distributed training might affect results, we run all SynFlow computation on CPU and subsequently train the pruned network on TPU.

**Double precision floats.** We compute the SynFlow scores using double precision floating point numbers. With single precision floating point numbers, the SynFlow activations explode on networks deeper than ResNet-44 (CIFAR-10) and ResNet-18 (ImageNet).

### D.3  NETWORKS AND DATASETS

Tanaka et al. consider the following settings for computer vision:

| SynFlow Name | Our Name | Dataset | GitHub | Replicated | Notes |
|---|---|---|---|---|---|
| VGG-11 | VGG-11 | CIFAR-10 | ✓ | ✓ | A shallower version of our VGG-16 network |
| VGG-11 | VGG-11 | CIFAR-100 | ✓ | ✗ | A shallower version of our VGG-16 network |
| VGG-11 | VGG-11 (Modified) | TinyImageNet | ✓ | ✗ | Modified for TinyImageNet |
| VGG-16 | VGG-16 | CIFAR-10 | ✓ | ✓ | Identical to our VGG-16 network |
| VGG-16 | VGG-16 | CIFAR-100 | ✓ | ✗ | Identical to our VGG-16 network |
| VGG-16 | VGG-16 (Modified) | TinyImageNet | ✓ | ✗ | Modified for TinyImageNet |
| ResNet-18 | ResNet-18 (Modified) | CIFAR-10 | ✓ | ✓ | Modified ImageNet ResNet-18; first conv is 3x3 stride 1; no max-pool |
| ResNet-18 | ResNet-18 (Modified) | CIFAR-100 | ✓ | ✗ | Modified ImageNet ResNet-18; first conv is 3x3 stride 1; no max-pool |
| ResNet-18 | ResNet-18 (Modified) | TinyImageNet | ✓ | ✓ | Modified ImageNet ResNet-18; first conv is 3x3 stride 1; no max-pool |

Table 6: The networks and datasets examined in the SynFlow paper (Tanaka et al., 2020)

Of the settings that we replicated, our unpruned network performance is as follows:

| Network | Dataset | Reported | Replicated | Notes |
|---|---|---|---|---|
| VGG-11 | CIFAR-10 | ~92% | 92.0% | Hyperparameters and augmentation are identical to ours |
| VGG-16 | CIFAR-10 | ~94% | 93.5% | Hyperparameters and augmentation are identical to ours |
| ResNet-18 (Modified) | CIFAR-10 | ~95% | 93.7% | Hyperparameters reported by Tanaka et al. (lr=0.01, batch size=128, drop factor=0.2) |
| ResNet-18 (Modified) | CIFAR-10 | — | 94.6% | Hyperparameters reported by Tanaka et al. (lr=0.2, batch size=256, drop factor=0.1) |
| ResNet-18 (Modified) | TinyImageNet | ~64% | 58.8% | Hyperparameters reported by Tanaka et al. (lr=0.01, batch size=128, epochs=100) |
| ResNet-18 (Modified) | TinyImageNet | — | 64% | Modified hyperparameters (lr=0.2, batch size=256, epochs=200) |

Table 7: Top-1 accuracy of unpruned networks as reported by Tanaka et al. (2020) and as replicated. Tanaka et al. showed plots rather than specific numbers, so reported numbers are approximate.

The VGG-11 and VGG-16 CIFAR-10 results are identical between our implementation and that of Tanaka et al..

We modified the standard ImageNet ResNet-18 from TorchVision to match the network of Tanaka et al.. We used the same data augmentation and hyperparameters on TinyImageNet, but accuracy was much lower (58.8% vs. 64%). By increasing the learning rate from 0.01 to 0.2, increasing the batch size to 256, and increasing the number of training epochs to 200, we were able to match the accuracy reported by Tanaka et al.. We also used the same data augmentation and hyperparameters on CIFAR-10, but accuracy was lower (95% vs. 93.6%). Considering that we needed different hyperparameters on both datasets, we believe that there is an unknown difference between our ResNet-18 implementation and that of Tanaka et al..

## D.4 RESULTS

Tanaka et al. compare to random pruning, magnitude pruning at initialization, SNIP, and GraSP at 13 sparsities evenly space logarithmically between 0% sparsity and 99.9% sparsity (compression ratio $10^3$). In Figure 7, we plot the same methods at sparsities between 0% and 99.9% at intervals of an additional 50% sparsity (e.g., 50% sparsity, 75% sparsity, 87.5% sparsity, etc.).

Tanaka et al. present graphs rather than tables of numbers. As such, in Figure 7, we compare our graphs (left) to the graphs from the SynFlow paper (right). On the VGG-style networks for CIFAR-10, our results look nearly identical to those of Tanaka et al. in terms of the accuracies of each method, the ordering of the methods, and when certain methods drop to random accuracy. The only difference is that SNIP encounters layer collapse in our experiments, while it does not in those of Tanaka et al.. Since these models share the same architecture and hyperparameters as in the SynFlow paper and the results look very similar, we have confidence that our implementation of SynFlow and the other techniques matches that of Tanaka et al..

Our ResNet-18 experiments look quite different from those of Tanaka et al.. The ordering of the lines is different, SynFlow is not the best performing method at the most extreme sparsities, and magnitude pruning does not drop to random accuracy. Considering the aforementioned challenges replicating Tanaka et al.'s performance on the unpruned ResNet-18, we attribute these differences to an unknown difference in our model or training configuration. Possible causes include different hyperparameters (which may cause both the original model and the pruned networks to perform differently). Another possible cause is a different initialization scheme: we initialize the $\gamma$ parameters of BatchNorm uniformly between 0 and 1 and use He normal initialization based on the fan-in of the layer (both PyTorch defaults) while Tanaka et al. initialize the $\gamma$ parameters to 1 and use He normal initialization based on the fan-out of the layer. Although these differences in the initialization scheme are small, they could make a substantial difference for methods that prune at initialization. This difference in results, despite the fact that both unpruned ResNet-18 networks reach the same accuracy on TinyImageNet, suggest that there may be a significant degree of brittleness, at least at the most extreme sparsities.

(The GraSP implementation of Tanaka et al. (2020) contains a bug: it uses batch normalization in evaluation mode rather than training mode, which may also explain some of the differences we see in GraSP performance, especially on Modified ResNet-18.)

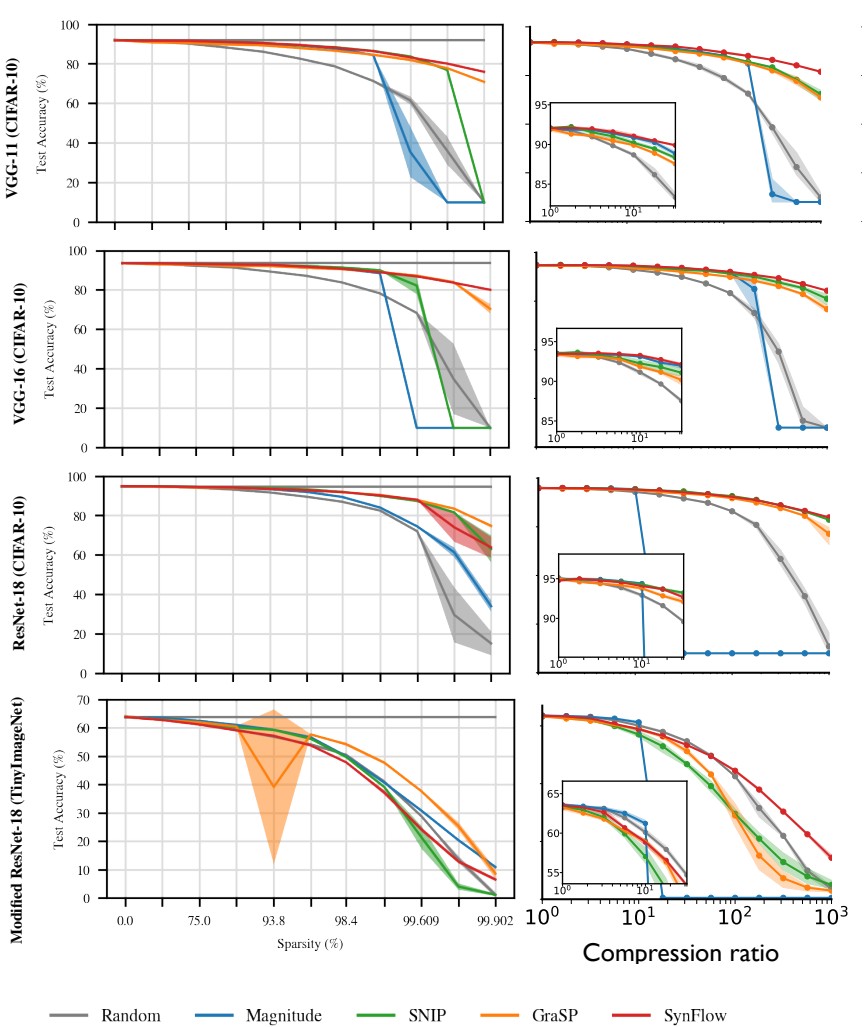

Figure 7: Synflow replication experiments.

### D.5 A CONNECTION BETWEEN SYNFLOW AND PATH NORM PRESERVATION

In this section, we elucidate a connection between SynFlow and a particular group-norm that has been studied in the setting of generalization bounds for neural networks.

For this subsection, fix a neural network architecture, represented by a directed, acyclic graph $(V, E)$ with neurons $V$ and connections $E \subset V \times V$. We will assume that the architecture is layered in the sense that each neuron is connected to every neuron in the layer before and after and no other neurons. A neural network $\mathcal{N} = (V, E, W)$ is an architecture and weights $W = (w_e)_{e \in E}$ associated to every connection. A *path* $p = (v_1, \ldots, v_{|p|}) \in V^*$ is a finite sequence of connected neurons, i.e., $(v_i, v_{i+1}) \in E$ for all $i$. Let $\mathcal{P}$ denote the set of all paths in the architecture. For a path $p = (v_1, \ldots, v_{|p|})$, we write $v \in p$ (resp., $e \in p$) for $v \in \{v_1, \ldots, v_{|p|}\}$ (resp., $e \in \{(v_1, v_2), \ldots, (v_{|p|-1}, v_{|p|})\}$).

Let $A = \{a_1, ..., a_D\} \subset V$ and $O = \{o_1, ..., o_K\} \subset V$ denote the input and output neurons, respectively. For $v, v' \in V$, let $\mathcal{P}(v)$ denote the set of all paths $p \in \mathcal{P}$ such that $v \in p$ and let $\mathcal{P}(v, v')$ denote the set of all paths in $\mathcal{P}$ starting at $v$ and ending at $v'$. For $e \in E$, let $\mathcal{P}(e)$ denote the set of all "complete" paths $p \in \mathcal{P}$ such that $e \in p$, where "complete" means that, for some $a \in A$ and $o \in O$, $p \in \mathcal{P}(a, o)$.

The $(\ell_{1,1})$ *path norm* $\mu(\mathcal{N})$ of a ReLU network $\mathcal{N}$ is the sum, over all paths $p$ from an input neuron to an output neuron, of the product of weights of each connection in $p$, i.e.,

$$\mu(\mathcal{N}) = \sum_{a \in A, \, o \in O} \sum_{p \in \mathcal{P}(a, o)} \left| \prod_{e \in p} w_e \right|.$$

The path norm is a special case of group norms studied by Neyshabur et al. (2015), towards bounding the Rademacher complexity of norm-bounded classes of neural networks with ReLU activations. As such, the path norm can be interpreted as a capacity measure for such networks. One of the key properties of the path norm is its invariance to a certain type of balanced rescalings of the weights that also leave the output of ReLU networks invariant. In particular, multiplying one layer of weights by $c$ while multiplying another layer by its reciprocal $1/c$ leaves the path norm invariant.

In order to relate the path norm to SynFlow, we relate the path norm to certain gradients. Consider the input equal to $(1, \ldots, 1)$ and let $\mathcal{N}$ be a network whose weights $W$ have all been made positive by setting each weight to its absolute value. In such a network, on input $(1, \ldots, 1)$, the preactivation value $f_v$ at every neuron $v \in V$ is positive, and so ReLU activations can be ignored/dropped. For every output neuron $o$, the gradient of $f_o$ with respect to a weight $w_e$ is

$$\frac{\partial f_o}{\partial w_e} = \sum_{p \in \mathcal{P}(e) \cap \mathcal{P}(o)} \prod_{e' \in p, \, e' \neq e} w_{e'}. \tag{5}$$

It it then straightforward to verify that, writing $E_\ell$ for all connections in the $\ell$'th layer,

$$\mu(\mathcal{N}) = \sum_{e \in E_\ell} |w_e| \sum_{o \in O} \frac{\partial f_o}{\partial w_e}.$$

We will now show that SynFlow can be seen to prune weights that maximally preserve the path norm locally. Taking the derivative of the path norm with respect to an individual weight gives

$$\frac{\partial \mu(\mathcal{N})}{\partial w_e} = \sum_{p \in \mathcal{P}(e)} \left| \prod_{e' \in p, \, e' \neq e} w_{e'} \right|.$$

Consider a ReLU network $\mathcal{N}$, and let $\mathcal{N}_{\neg e}$ be the subnetwork with all the paths through connection $e$ removed. In other words, $\mathcal{N}_{\neg e}$ is the network after pruning the weight $w_e$. We can approximate how removing the weight $w_e$ determines the path norm of the resulting subnetwork $\mathcal{N}_{\neg e}$. In the first-order Taylor series approximation, removing the weight $w_e$ decreases the path norm by

$$|w_e| \frac{\partial \mu(\mathcal{N})}{\partial w_e} = \sum_{p \in \mathcal{P}(e)} \left| \prod_{e' \in p} w_{e'} \right|,$$

which, by (5), can be rewritten as

$$|w_e| \sum_{o \in O} \frac{\partial f_o}{\partial w_e}.$$

We see that (D.5) is identical to SynFlow's weight score function. In particular, the highest-scoring weight according to SynFlow is the weight such that, an infinitesimal perturbation has the smallest impact on the path norm.

# E  BASELINES

In Figure 8, we show the four baseline methods to which we compare SNIP, GraSP, and SynFlow: random pruning at initialization, magnitude pruning at initialization, magnitude pruning after training, and lottery ticket rewinding. Lottery ticket rewinding reaches different accuracies depending on the iteration $t$ to which we set the state of the pruned network. We use $t = 1000, 2000, 1000$, and 6000 for ResNet-20, VGG-16, ResNet-18, and ResNet-50; as shown in Figure 6, these are the iterations where accuracy improvements saturate.

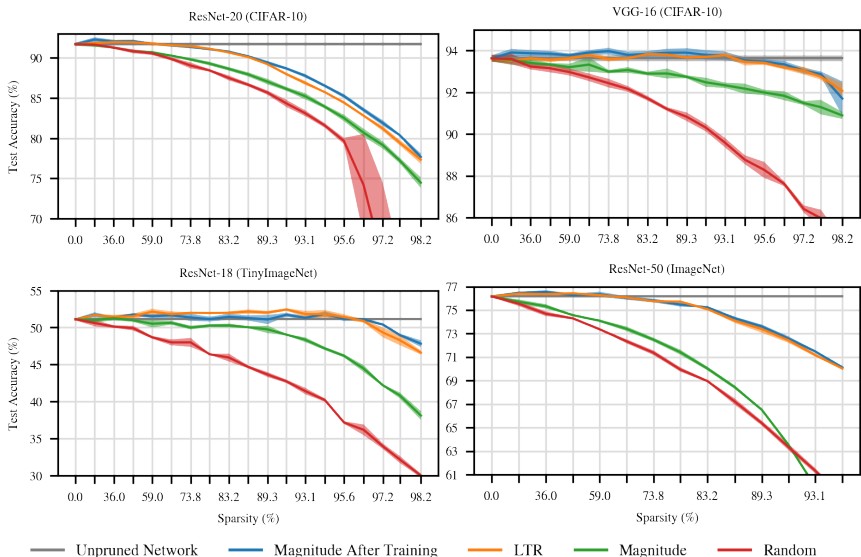

Figure 8: Accuracy of the baseline methods.

## F    ABLATIONS AFTER INITIALIZATION

In this appendix, we perform the ablations (Section 5) on the pruning methods when applied *after* initialization (Section 6). Our goal is a continuation of Section 6: to study whether the pruning methods become more sensitive to the ablations as pruning occurs later in training.

### F.1    ABLATIONS AT THE END OF TRAINING

In Figure 9, we perform the ablations from Section 5 when pruning at the end of training. This figure parallels Figure 4, where we performed these ablations when pruning at initialization.

**Magnitude.** Our results in Figure 9 confirm the well-known result that shuffling or reinitializing subnetworks found via magnitude pruning after training causes accuracy to drop (Han et al., 2015; Frankle et al., 2020a). On ResNet-20, shuffling and reinitializing perform similarly; they match full accuracy until 49% sparsity vs. 73.8% for the unmodified network. On VGG-16 and ResNet-18, random pruning performs better initially, after which reinitializing overtakes it at higher sparsities.

**SNIP.** On SNIP, there are smaller differences in accuracy between the unmodified network and the ablations. On ResNet-20, the unmodified network performs slightly better. On VGG-16, it performs approximately as well as when randomly shuffled. On ResNet-18, there are more substantial differences. These results indicate that SNIP is not inherently robust to these ablations, but rather that the point in training at which SNIP is applied plays a role. Note that, at the highest sparsities on ResNet-20 and VGG-16, SNIP did not consistently converge, leading to the large error bars.

**GraSP.** On GraSP, the ablations have a limited effect on the performance of the network, similar to pruning at initialization. On ResNet-20, the unmodified networks and the ablations perform the same. On VGG-16, the shuffling ablation actually outperforms the unmodified network (which performs the same as when reinitialized) at lower sparsities. On ResNet-18, all three experiments perform similarly at lower sparsities, and shuffling performs lower at extreme sparsities.

**SynFlow.** On SynFlow, the ablations do affect performance, but in different ways on different networks. On ResNet-20, shuffling improves accuracy; on VGG-16 and ResNet-18 it decreases accuracy at higher sparsities.

### F.2    ABLATIONS AFTER INITIALIZATION

In Figure 10, we perform the ablations at all points in training at a single sparsity for each network. This figure shows the ablations corresponding to Figure 6 in Section 6.

**Magnitude.** Magnitude pruning becomes sensitive to the ablations early in training (after iteration 5,000 for ResNet-20, 10,000 for VGG-16, and 4,000 for ResNet-18). The performance of the ablations does not improve after the very earliest part of training; the performance without the ablations continues improving after this point, while the performance of the ablations remains the same after this point. On VGG-16, shuffling outperforms reinitialiation, suggesting the initialization matters more tan the structure of the sparse network. On ResNet-18, the reverse is true.

**SNIP.** The behavior of SNIP is very similar to the behavior of magnitude.

**GraSP.** GraSP improves little when pruning after initialization. It only sensitive to the shuffling ablation on VGG-16 and ResNet-18, and the changes in accuracy under this ablation are small.

**SynFlow.** SynFlow is sensitive to the ablations, but in different ways on different networks. On ResNet-20 and VGG-16, the accuracy under shuffling improves alongside the accuracy of unmodified SynFlow. On ResNet-18, the accuracy under shuffling is lower and does not improve. In all cases, accuracy is lower under reinitialization.

### F.3    SUMMARY

Overall, these results support our hypothesis that it may be especially difficult to prune in a connection-specific or initialization-sensitive manner at initialization. At initialization, magnitude pruning, SNIP, and SynFlow maintain or improve upon their accuracy under random shuffling and reinitialization. After initialization, shuffling and reinitialization hurt accuracy to varying degrees.

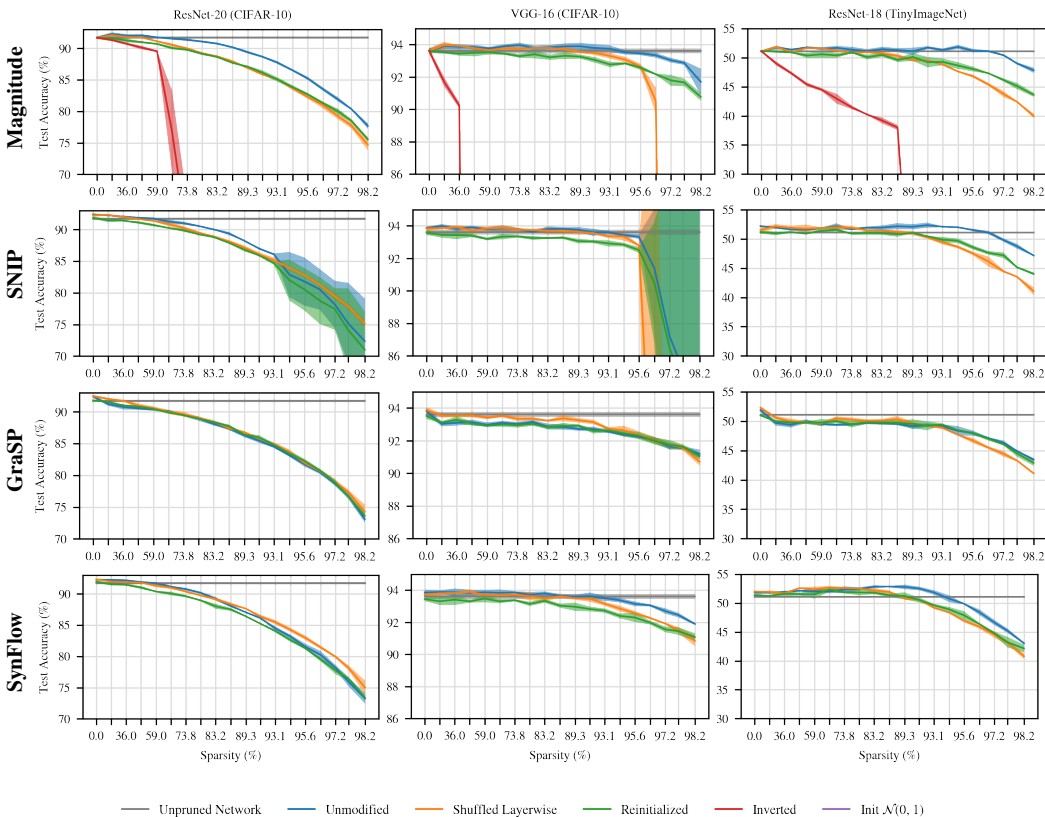

Figure 9: Ablations on subnetworks found by applying magnitude, SNIP, GraSP, and SynFlow after training. (We did not run this experiment on ResNet-50 due to resource limitations.)

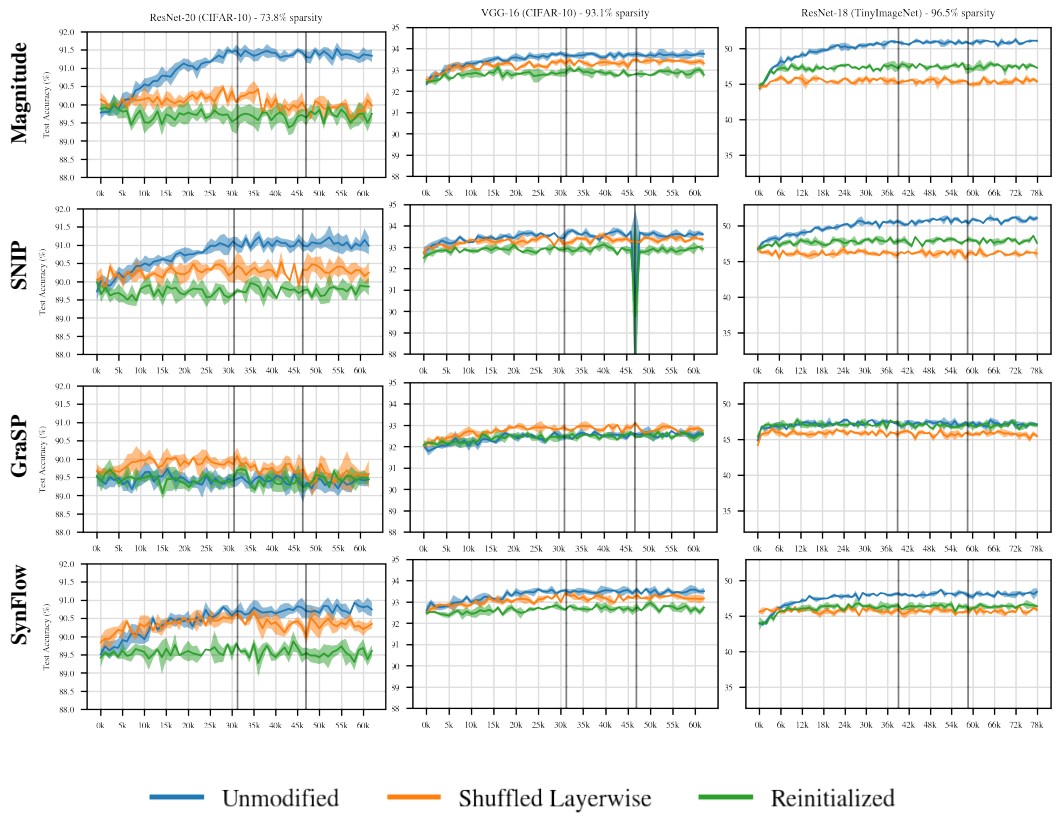

Figure 10: Accuracy of early pruning methods and ablations when pruning at the iteration on the x-axis. Sparsities are the highest matching sparsities. Vertical lines are iterations where the learning rate drops by 10x.

# G    ITERATIVE SNIP

In the main body of the paper, we consider SNIP as it was originally proposed by Lee et al. (2019), pruning weights from the network in one shot. Recently, however, de Jorge et al. (2020) and Verdenius et al. (2020) have proposed variants of SNIP in which pruning occurs iteratively (like SynFlow). In iterative variants of SNIP, the SNIP scores are calculated, some number of weights are pruned from the network, and the process repeats multiple times with new SNIP scores calculated based on the pruned network. In this appendix, we compare one-shot SNIP (the same experiment as in the main body of the paper) and iterative SNIP (in which we prune iteratively over 100 iterations, with each iteration going from sparsity $s^{\frac{n-1}{N}}$ to sparsity $s^{\frac{n}{N}}$—the same strategy as we use for SynFlow).

Figure 11 below shows the performance of SNIP (red) and iterative SNIP (green) at the sparsities we consider. At these sparsities, making SNIP iterative does not meaningfully alter performance. It is possible that making SNIP iterative matters most at the especially extreme sparsities studied by de Jorge et al. (2020) and Tanaka et al. (2020) rather than at the matching and relatively less extreme sparsities we focus on in this paper.

Figure 12 shows the shuffling and reinitialization ablations for SNIP (top) and iterative SNIP (bottom). In both cases, the ablations do not meaningfully affect accuracy.

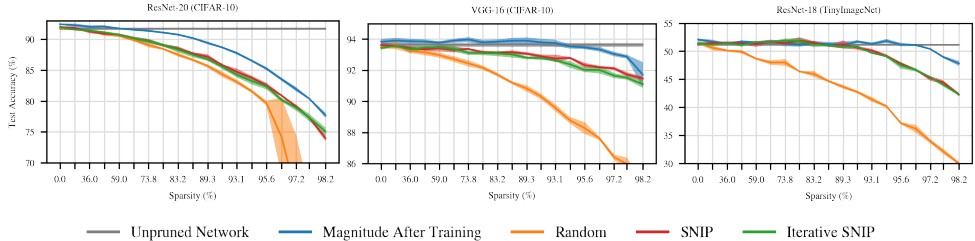

Figure 11: A comparison of SNIP and iterative SNIP (100 iterations).

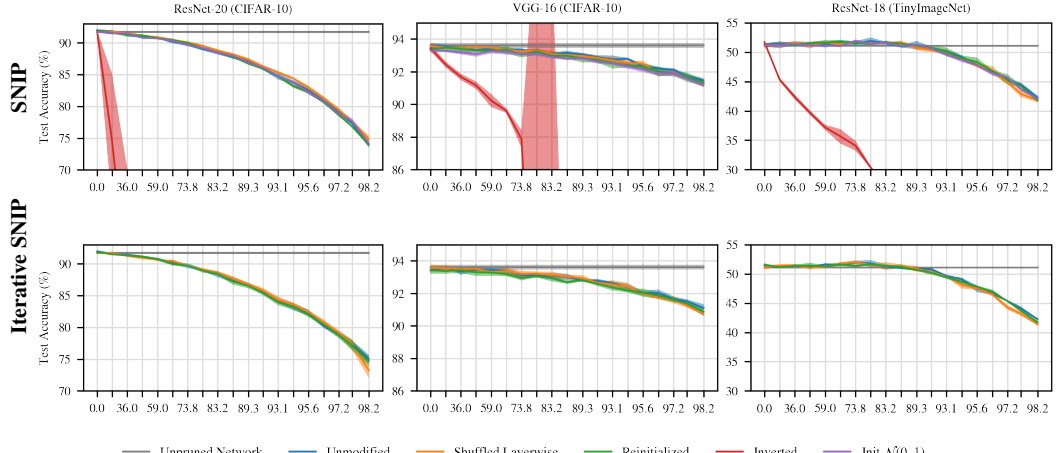

Figure 12: A comparison of the ablations for SNIP (from the main body, Figure 4) and for iterative SNIP. (ResNet-50 is not included due to computational limitations.)

# H  VARIANTS OF GRASP

In Figure 13, we show three variants of GraSP: pruning weights with the highest scores (the version of GraSP from Wang et al.), pruning weights with the lowest scores (the inversion experiment from Section 5), and pruning weights with the lowest magnitude GraSP scores (our proposal for an improvement to GraSP as shown in Figure 14). This figure is intended to make the comparison between these variants clearer; figure 4 is too crowded for these distinctions to be easily visible.

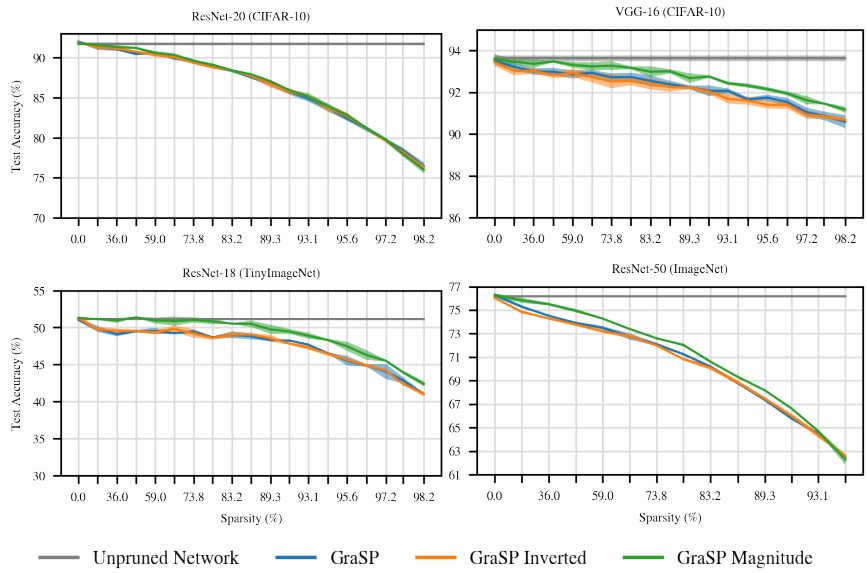

Figure 13: Accuracy of three different variants of GraSP

# I COMPARISONS TO IMPROVED GRASP AND SYNFLOW

In Figure 14 below, we compare the early pruning methods with our improvements to GraSP and SynFlow. This figure is identical to Figure 3, except that we modify GraSP to prune the weights with the lowest-magnitude GraSP scores (Appendix H) and we modify SynFlow to randomly shuffle the per-layer pruning masks after pruning (Section 5).

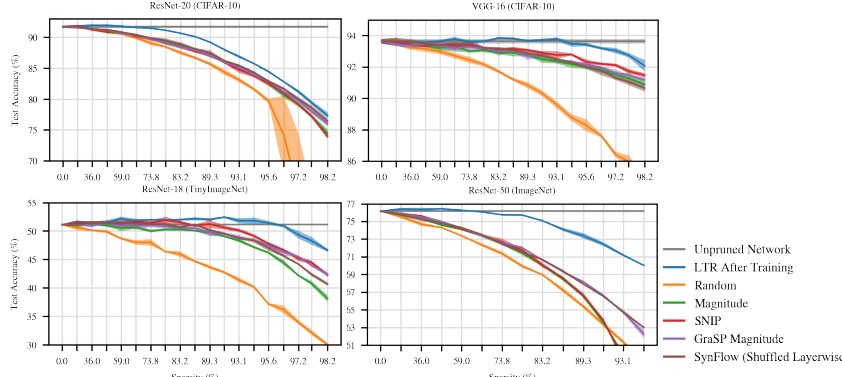

Figure 14: The best variants of pruning methods at initialization from our experiments. GraSP and SynFlow have been modified as described in Section 5.

## J    LAYERWISE PRUNING PROPORTIONS

In Figure 15 on the following page, we plot the per-layer sparsities produced by each pruning method for at the highest matching sparsity. Each sparsity is labeled with the corresponding layer name; layers are ordered from input (left) to output (right) with residual shortcut/downsample connections placed after the corresonding block. We make the following observations.

**Different layerwise proportions lead to similar accuracy.** At the most extreme matching sparsity, the early pruning methods perform in a relatively similar fashion: there is a gap of less than 1, 1.5, 2.5, and 1 percentage point between the worst and best performing early pruning methods on ResNet-20, VGG-16, ResNet-18, and ResNet-50. However, the layerwise proportions are quite different between the methods. For example, on ResNet-20, SynFlow prunes the early layers to less than 30% sparsity, while GraSP prunes to more than 60% sparsity. SNIP and SynFlow tend to prune later layers in the network more heavily than earlier layers, while GraSP tends to prune more evenly.

On the ResNets, the GraSP layerwise proportions most closely resemble those of magnitude pruning after training, despite the fact that GraSP is not the best-performing method at the highest matching sparsity on any network.

These results are further evidence that determining the layerwise proportions in which to prune (rather than the specific weights to prune) may not be sufficient to reach the higher performance of the benchmark methods. The early pruning methods produce a diverse array of different layerwise proportions, yet performance is universally limited.

**Skip connections.** When downsampling the activation maps, the ResNets use 1x1 convolutions on their skip connections. On ResNet-20, these layer names include the word `shortcut`; on ResNet-18 and ResNet-50, they include the word `downsample`. SynFlow prunes these connections more heavily than other parts of the network; in contrast, all of the other methods prune these layers to similar (ResNet-50) or much lower (ResNet-20 and ResNet-18) sparsities than adjacent layers. On ResNet-50, SynFlow entirely prunes three of the four downsample layers, eliminating the residual part of the ResNet for the corresponding blocks.

**The output layer.** All pruning methods (except random pruning) prune the output layer at a lower rate than the other layers. These weights are likely disproportionately important to reaching high accuracy since there are so few connections and they directly control the network outputs.

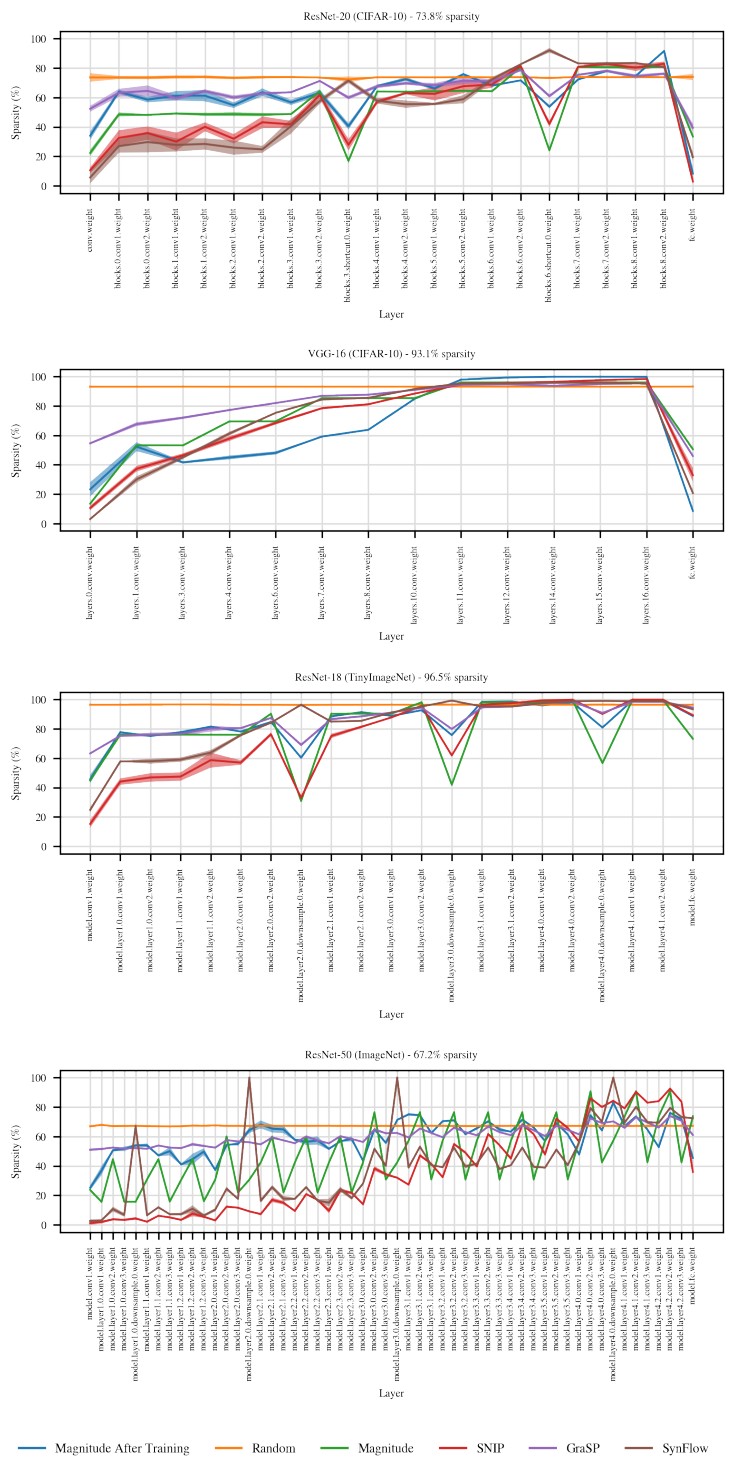

Figure 15: Per-layer sparsities produced by each pruning method at the highest matching sparsity.

## K   LeNet-300-100 for MNIST

In this appendix, we show selected experiments from the main body of the paper and the appendices on the fully-connected LeNet-300-100 for MNIST. We did not include this setting in the main body of the paper because results on MNIST often do not translate to other, larger-scale settings. The reason why we include this experiment here is to evaluate our ablations in the context of a fully-connected network. Unlike a convolutional network, each weight of the first layer of a fully-connected network is tied to a specific pixel of the input, so it is possible that randomly shuffling the pruned weights may have a different effect in this setting.

### K.1   Figure 3

For the standard versions of the methods, SNIP performs best and magnitude pruning at initialization performs worst, although all of the methods perform similarly at lower sparsities. Compared to the larger convolutional networks, there is little distinction between the performance of magnitude pruning after training and the early pruning methods—just a small fraction of a percent. This result is further evidence that LeNet-300-100 on MNIST is not representative of larger-scale settings.

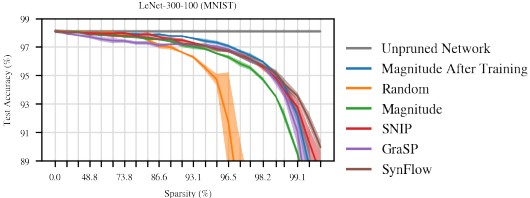

### K.2   Figure 4

The ablations vary by network. For magnitude pruning, shuffling and reinitializing do not affect accuracy. For SNIP, shuffling and reinitializing do not affect accuracy except at the higher sparsities. For GraSP, shuffling and reinitializing increase accuracy at lower sparsities and hurt accuracy at higher sparsities; unlike the other networks, inverting GraSP does hurt accuracy. Finally, SynFlow is unaffected by shuffling at lower sparsities and is unaffected by reinitializing at all sparsities.

Recall that the motivating research question for this section was whether random shuffling would have a different effect on a fully-connected network in which each connection in the first hidden layer is specific to a particular pixel of the input. For this reason, one possible hypothesis is that random shuffling will hurt accuracy in a manner that it did not for the convolutional networks.

These ablations show that randomly shuffling only hurts accuracy (compared to the unmodified pruning strategies) at the highest sparsities if at all. At these sparsities, the network has been pruned to the point where accuracy begins to plummet. We conclude that the hypothesis about the effect of shuffling on a fully-connected network does hold, but only at the most extreme sparsities. For comparison, shuffling affects the accuracy of magnitude pruning after training (Figure 9 below, upper left plot) at much lower sparsities.

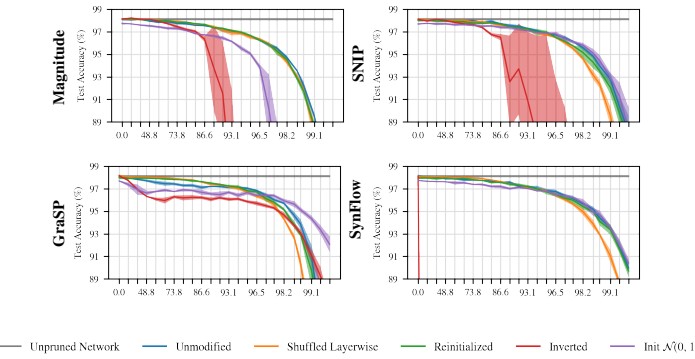

### K.3   FIGURE 5

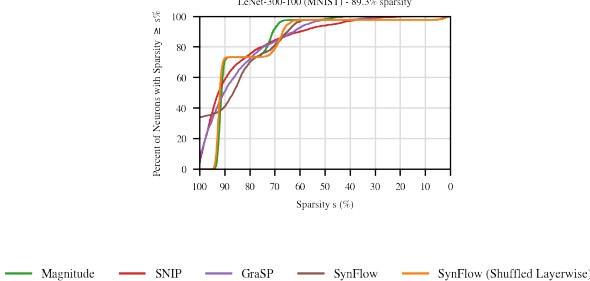

### K.4   FIGURE 8

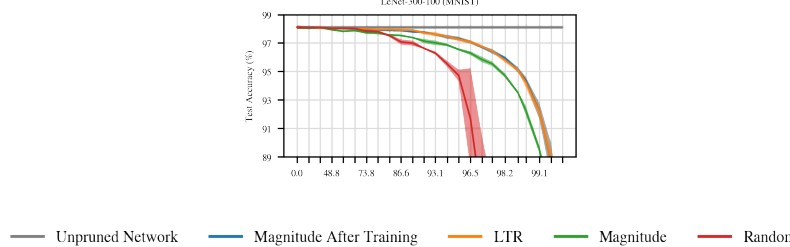

### K.5   FIGURE 9

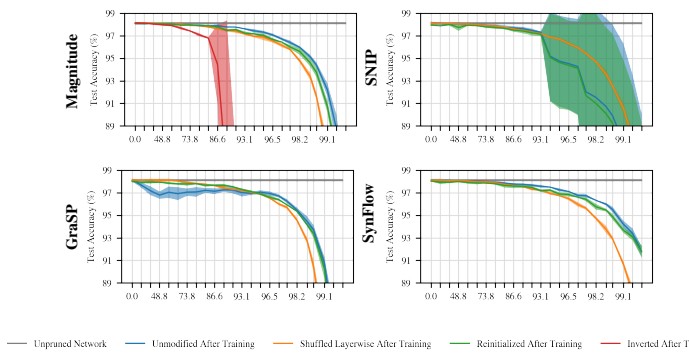

### K.6   FIGURE 11

Unlike the convolutional networks, iterative SNIP improves accuracy at the most extreme sparsities. It is possible that it will have the same effect for the convolutional networks at more extreme sparsities than wstudy in this paper.

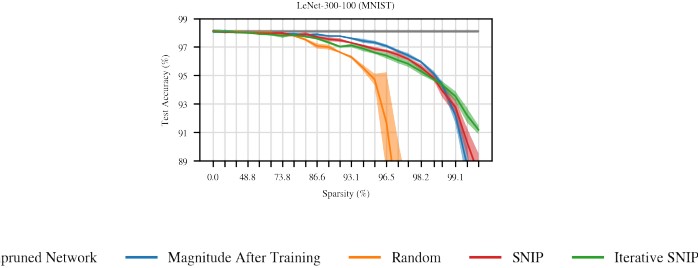

## K.7   FIGURE 13

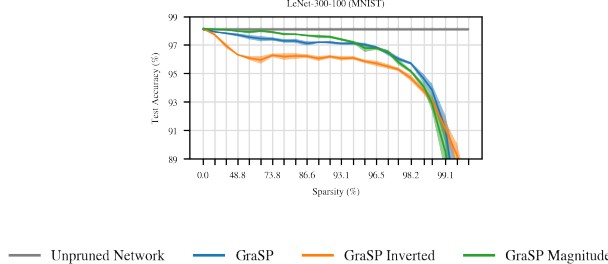

## K.8   FIGURE 15

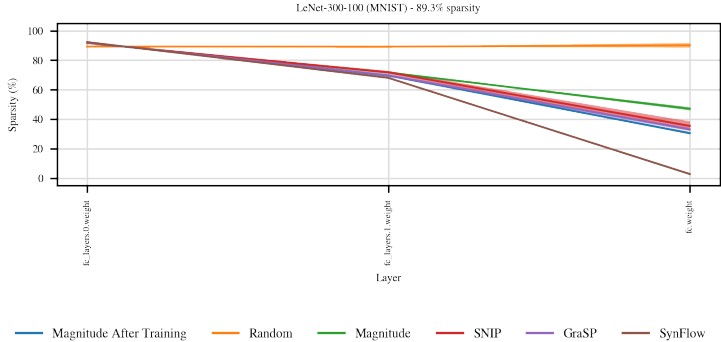

## L    MODIFIED RESNET-18 FOR TINYIMAGENET

In this appendix, we show the experiments from the main body of the paper on the modified version of ResNet-18 on TinyImageNet modeled after the setting used by Tanaka et al. (2020) to evaluate SynFlow. This is the same configuration as we describe in Appendix A.

We include this experiment because the TinyImageNet benchmark is less standardized than other benchmarks the we include in the paper (e.g., ResNets, VGGs, CIFAR-10, and ImageNet). We use this appendix to validate that, for a very different realization of the TinyImageNet benchmark (modified ResNet-18, different augmentation, and higher accuracy), our main findings hold.

### L.1    FIGURE 3

For the standard versions of the early pruning methods, magnitude pruning at initialization performs best at lower sparsities and GraSP performs best at higher sparsities. Surprisingly, SynFlow is no better than random pruning. At the higher accuracy this variant of TinyImageNet reaches, magnitude pruning after training is matching only at much lower sparsities.

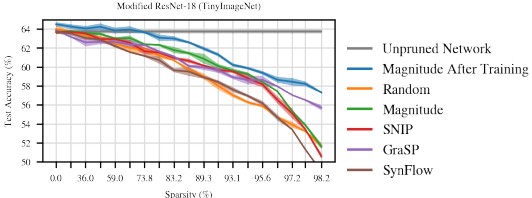

### L.2    FIGURE 4

As in the main body of the paper, we find that all methods maintain or improve upon their accuracy when randomly shuffling (Figure 4). SynFlow shows dramatic improvements, and SNIP improves as well. All methods maintain their performance when randomly reinitializing. Only magnitude pruning degrades in performance when changing the initialization distribution to have a fixed variance. Finally, GraSP maintains its performance when inverting, while the other methods degrade in performance.

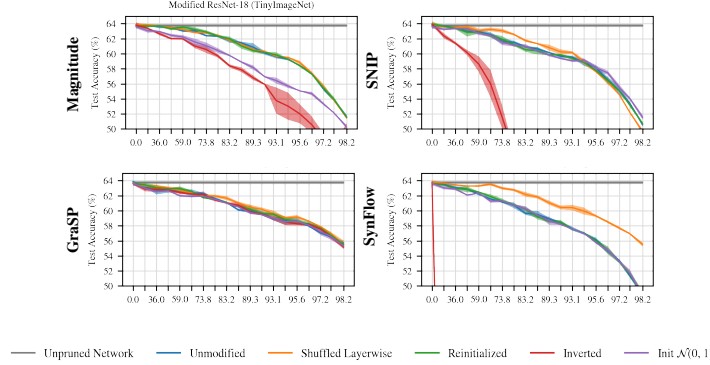

### L.3   FIGURE 5

As in the main body of the paper, SynFlow leads to neuron collapse, and randomly shuffling reduces the extent of this neuron collapse (Figure 5).

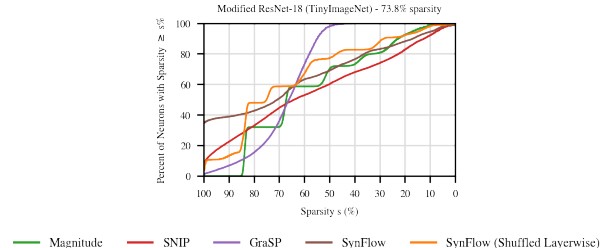

### L.4   FIGURE 8

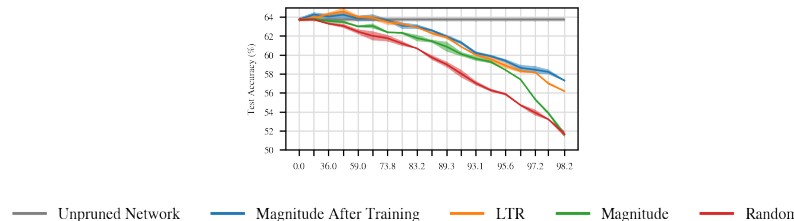

### L.5   FIGURE 9

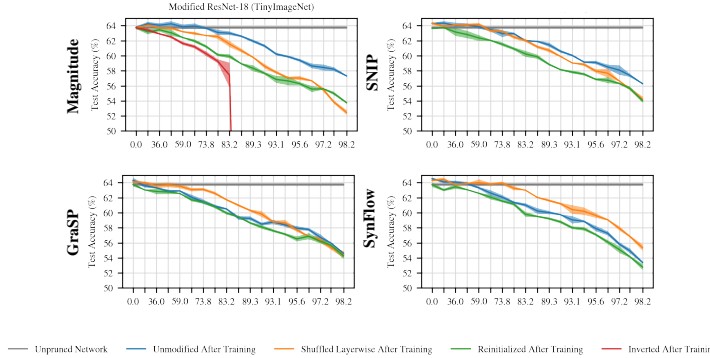

### L.6   FIGURE 13

Pruning the weights with the lowest-magnitude GraSP scores does not change performance, whereas it improves performance in some cases in the main body of the paper.

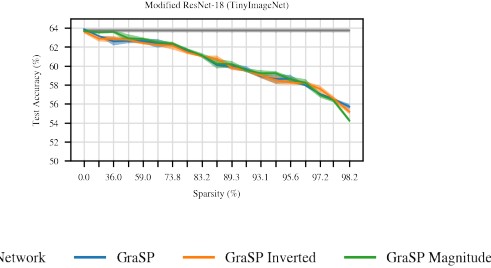

## L.7    FIGURE 15

As in Appendix J, SynFlow prunes skip connection weights with a higher propensity than other methods (Figure 15).

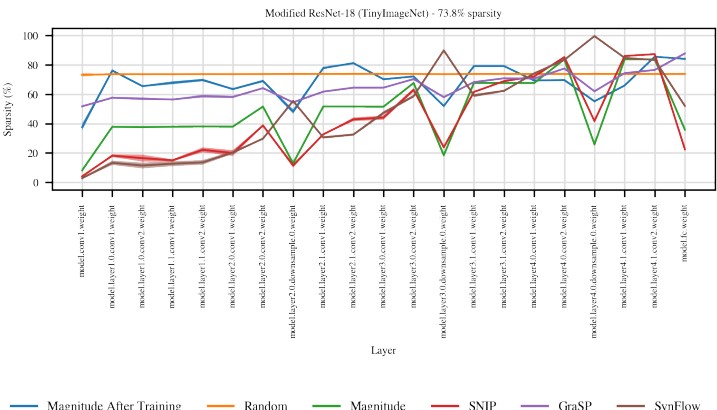

## M   COMPARING THE ACTUAL AND EFFECTIVE SPARSITY OF RANDOMLY SHUFFLED NETWORKS

In this appendix, we evaluate the following hypothesis:

*It is possible that the early pruning methods leave some weights unpruned but disconnected, which means the "effective sparsity" (the sparsity when also eliminating disconnected weights) of these networks is higher than the "actual sparsity" (the sparsity when taking into account only connections that are explicitly pruned). Furthermore, when randomly shuffling, some of these disconnected weights may become reconnected. As such, random shuffling might appear to maintain performance simply because it has a lower effective sparsity than the unmodified pruning technique.*

To evaluate this hypothesis, we compare the effective sparsities and actual sparsities for the early pruning methods with and without randomly shuffling.

**Methods.** To determine which weights are disconnected, we use the following procedure:

1. Set each weight in the network to 1 if it is unpruned or 0 if it is pruned.
2. Forward-propagate a single example comprising all 1's.
3. Compute the sum of the logits.
4. Compute the gradients with respect to this sum.
5. Prune any unpruned weight with a gradient of 0. Since these weights did not receive any gradient, they are disconnected from the output of the network.

**Evaluation.** For each actual sparsity, we compute the ratio of $\frac{\text{effective parameters in the shuffled network}}{\text{effective parameters in the unmodified network}}$. This allows us to determine the extent to which the shuffled network is larger or smaller than the unmodified pruned network.

**Results.** In Figure 16, we show this ratio for all pruning methods and all sparsities on our four main networks. In the table below, we summarize our results at the highest matching sparsity and highest sparsity that we consider.

| Network | Pruning Method | Highest Matching Sparsity | | Highest Sparsity We Consider | |
| --- | --- | --- | --- | --- | --- |
| | | Sparsity | Ratio | Sparsity | Ratio |
| ResNet-20 (CIFAR-10) | Magnitude | 73.8% | 1.0000 | 98.2% | 1.0016 |
| | SNIP | | 1.0006 | | 1.0353 |
| | GraSP | | 1.0000 | | 1.0197 |
| | SynFlow | | 1.0000 | | 0.9921 |
| VGG-16 (CIFAR-10) | Magnitude | 93.1% | 1.0000 | 98.2% | 0.9998 |
| | SNIP | | 1.0007 | | 1.0058 |
| | GraSP | | 1.0003 | | 1.0036 |
| | SynFlow | | 1.0000 | | 1.0001 |
| ResNet-18 (TinyImageNet) | Magnitude | 96.5% | 1.0003 | 98.2% | 1.0000 |
| | SNIP | | 1.0026 | | 1.0227 |
| | GraSP | | 1.0011 | | 1.0030 |
| | SynFlow | | 1.0001 | | 1.0001 |
| ResNet-50 (ImageNet) | Magnitude | 67.2% | 1.0000 | 94.5% | 0.9979 |
| | SNIP | | 1.0003 | | 1.0016 |
| | GraSP | | 1.0000 | | 1.0009 |
| | SynFlow | | 1.0000 | | 1.0001 |

At the highest matching actual sparsity, the effective sparsity of the shuffled network was larger by a negligible amount: at most 1.0026x (i.e., 0.26%) larger than the unmodified network and in most cases closer to 1.0005x (i.e., 0.05%) larger. At more extreme sparsities, this ratio was higher but still small: at most 1.0353x (i.e., 3.5% larger) and in most cases closer to 1.005x (i.e., 0.5%) larger. We conclude that, although random shuffling does often restore some disconnected parameters, the actual fraction of parameters it restores is minuscule at and beyond the matching sparsities we focus on. These differences are so small that the effective sparsities round to the same values as the actual sparsities on the x-axis labels of our plots.

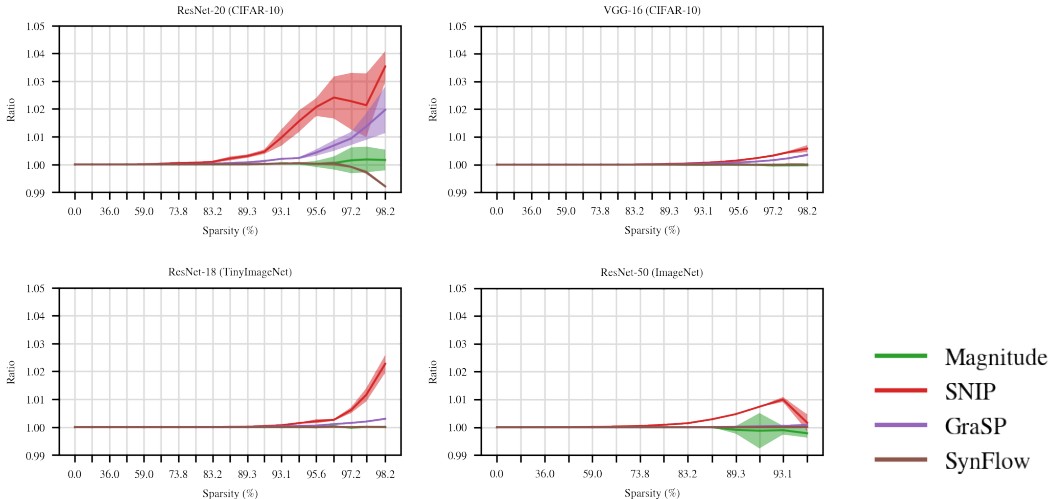

Figure 16: For each actual sparsity (x-axis), the ratio of the effective parameter-count of the randomly shuffled ablation to the effective parameter-count of the network pruned with the unmodified pruning method. Note that we have zoomed into the range of ratios between 0.99 and 1.05.

