# OpenReview forum: "Pruning Neural Networks at Initialization: Why Are We Missing the Mark?"
_ICLR.cc/2021/Conference — ICLR 2021 Poster_

### Official Review · AnonReviewer2 · 2020-10-28
**An excellent tour-de-force review which carefully understands, investigates and scrutinizes recent work in neural network pruning from scratch**

**Rating:** 9
**Confidence:** 5

**Review:**

A recent trend of 'pruning at initialization' in neural network pruning has left me baffled. It's counter-intuitive that neural networks can be pruned at initialisation, improving results for the training done thereafter. Nitpicking semantics, one could hardly even call this a pruning technique, since there is no a-priori knowledge of the dataset distilled in the network. Perhaps it's more aptly referred to as a method of sparse initialisation methods.
The authors of this paper seem to have had the same doubts when it comes to the pruning at initialization literature, and put the magnifying glass on these methods that have recently been published. In an extensive and comprehensive study, they show that pruning at initialization methods do naught but set per-layer sparsity rates, where the sparse initialization might as well have been random.

Writing these types of papers is important. It's essentially a survey paper, reproducing results from other papers, and testing their claims. In moving forward in this field, this type of work is crucial in filtering out the sense from the nonsense. Although the paper does not provide any new shiny optimization method, or stellar new GAN with a funny name, I highly laude the authors for working on this survey, and I believe the impact on the model efficiency community of this work is significant.

Now on to the nitty-gritty of the paper. The key argument that shuffling the weights within a layer, essentially functioning like a re-initialization with a given per-layer sparsity ratio, is solid, convincing and damning. The inversion arguments equally so. The only qualm I have with the results are that there are not more on different architectures... but I hardly think that's necessary to make the point. If these methods were to work as intended, it should show on the common computer vision architectures we work with.

The paper is well-written and well-structured. Clear and concise, with an extensive appendix highlighting more background information, and showing that the authors know the field very well. They covered every angle I could think of to put a crowbar in the paper and pry open some problems.

I would have liked to see a discussion on why we would even want to prune at initialization, because the arguments in the paper only really come to light if we consider the utility of the methods. Sure, pruning at initialization is worse than pruning, but perhaps there are reasons to prune at initialization anyway. Are we looking at this as a research exercise? A quest to improve our understanding of networks? Is it done for sparse training? If it's the latter, a discussion on this, and perhaps a comparison to other sparse training methods would be in order.

---

> ### Author Response · Authors · 2020-11-19
> **Author Response**
>
> We are grateful that the reviewer appreciated our purpose and the potential significance of our work. We have responded to the reviewer’s comments in-line below.
>
> ---
>
> _The only qualm I have with the results are that there are not more on different architectures._
>
> When it comes to networks, datasets, and scale, our paper is the most comprehensive study ever conducted on the topic of pruning neural networks at initialization, and we pushed our resources and our budget to the limit in order to make this happen.
>
> In the main body of our paper alone, we present results from training more than 11,000 networks. This includes training about 600 networks (14 sparsities) on ImageNet; for comparison, the GraSP paper (Wang et al.) is the only paper on pruning at initialization that includes any results on ImageNet, and it contains just 36 networks (3 sparsities) trained on the task.
>
> Since our purpose was to rigorously evaluate and examine these techniques, we chose to focus our limited resources on fewer settings in depth rather than more settings in a shallower manner. The papers introducing the methods we look at primarily (SNIP) or only (GraSP, SynFlow) studied image classification; our work therefore assesses these methods in their original domains. We were cognizant that reviewers or authors might consider it unfair if we included other computer vision tasks or natural language processing tasks for which these techniques were not designed.
>
> Although we agree that any empirical paper is stronger if it includes more settings, we believe the current experiments are convincing and comprehensive and we lack the resources to add more large-scale settings.
>
> ---
>
> _I would have liked to see a discussion on why we would even want to prune at initialization._
>
> We are in the process of updating the discussion section to provide an overview for possible motivations for pruning at initialization. We will post another update once this is complete.
>
> ---
>
> _Perhaps a comparison to other sparse training methods would be in order._
>
> The most prominent alternative class of sparse training methods are those that dynamically change the sparsity pattern throughout training (e.g., Mocanu et al., Dettmers & Zettlemoyer, Evci et al.). This is an exciting alternative direction for training sparse networks from the start, however, they are beyond the scope of this work. Our goal was to study the growing literature on pruning at initialization and to highlight the apparent difficulty of doing so; dynamic pruning methods begin with a sparse network, but they mainly prune after initialization, so they do not fall into this category. Many of our experiments do not make sense in the context of dynamic pruning, for example reinitializing after pruning (Section 5) and pruning only at a specific point in training (Section 6).

---

> > ### Author Response · Authors · 2020-11-24
> > **Paper is Updated**
> >
> > We wanted to notify you that we have posted our revised version of the paper. As promised, we have added a paragraph to the discussion explaining the various justifications for pruning at initialization. We have also made many changes throughout to address comments from other reviewers.

---

### Official Review · AnonReviewer1 · 2020-10-28
**Not convincing enough to support the claim**

**Rating:** 4
**Confidence:** 4

**Review:**

This is an empirical work assessing some of the recent works of early pruning methods in an effort to understand why early pruning methods perform worse than pruning after training methods. The paper is written very well, and some of the findings are interesting leaving important questions to address. Some major concerns remain though.

The paper proposes to achieve the following goal: to understand why early pruning methods perform worse than pruning after training methods.To achieve this, what the paper chooses to do is basically to measure their final performances and compare them to each other, and then to check for some ablation studies.

Obviously, the results (Figure 3) show which performs better than the other (i.e. pruning after training > pruning early), meaning that this result itself in terms of the final performance does not suffice to prove any cause or effect to explain the behaviors in comparison aimed to understand.

In the ablation studies (Section 5), the paper appears to be focused on studying and debunking the early pruning methods rather than finding *why* pruning after training is better than pruning early. For example, the result of random shuffling (early pruning methods for finding layerwise proportions) and the failure of GraSP when inverted are quite interesting, but they remain empirical without contributing directly to fulfill the purpose; some of the conclusions drawn from the ablation studies are not convincing (e.g., lack of specificity or insensitivity to initialization may limit performance); some of the claims are not expected to generalize either (e.g., shuffling to fully-connected layers, initializing with different standard deviations of the normal distribution).

In fact, an intuitive ablation setting is to test for the effect of training with the same saliency measure, as done in Section 6, but unfortunately the result out of this seems to be unhelpful for figuring out the potential cause of poor performances for pruning early methods. The results however show that early pruning methods improve after some training, indicating that training to some degree before pruning helps to increase the performance of the pruned model. One of the potentially many reasons that pruning early methods is underperformed by pruning after training methods could merely be the fact that the former is not trained before it gets pruned as opposed to the latter yielding the observed performance gap quite obviously. Anyway, the fact that they (after some training) still do not perform better than LTR (which requires oracle information) does not contribute to proving anything for why pruning early methods is underperformed by pruning after training either.

In short, this work does not explain "why all methods fall short of magnitude pruning after training" in a convincing manner, and the contributions are considered not quite significant overall.

---

> ### Author Response · Authors · 2020-11-24
> **Author Response (Part 2)**
>
> _Some of the claims are not expected to generalize either (e.g., shuffling to fully-connected layers, initializing with different standard deviations of the normal distribution)._
>
> In Appendix K.2, we specifically address shuffling fully-connected layers. Our claims about the ablations indeed generalize for the range of sparsities we consider.
>
> We study one modified initialization scheme (N(0, 1)) for the purposes of exploring how the pruning techniques behave when the initialization variance does not scale with layer sizes in the way that He initialization does. Our goal is an existential result (SNIP, GraSP, and SynFlow can “maintain accuracy in a case where the initialization is not informative for pruning in this way”) rather than a universal result (e.g., that these methods work with all possible initializations), so we do not believe it is relevant whether this result generalizes to other initialization schemes. We have modified the text (“even when the initialization is not informative” -> “in a case where the initialization is not informative”) to ensure this point is clear.
>
> ---
>
> _[The] test for the effect of training with the same saliency measure [in Section 6]...does not contribute to proving anything for why early pruning methods is underperformed by pruning after training._
>
> Our goal in performing this ablation was to determine whether the behaviors we observe earlier in the paper (lower accuracy than pruning after training and insensitivity to the ablations) are intrinsic to these particular pruning heuristics or whether they are specific to using these heuristics at initialization. We find that (with the possible exception of GraSP) these behaviors occur at initialization but not after a small amount of training. We do so by showing that the subnetworks that SNIP, SynFlow and magnitude find after initialization are able to reach higher accuracy (Section 6) and are sensitive to the ablations (Appendix F.2). This result means that our observations are specific to initialization and, more broadly, it raises a question about whether pruning at initialization inherently entails lower accuracy and insensitivity to the ablations.
> We have updated the description of this experiment in Sections 1 and 6 to improve clarity.
> Finally, although we show that these methods encounter particular difficulties at initialization, we admittedly do not identify the underlying properties of the network at initialization that lead to these difficulties (i.e., “why” these behaviors occur at initialization in particular). We have added a paragraph to the discussion explicitly mentioning this open question.
>
> ---
>
> _One of the potentially many reasons that pruning early methods is underperformed by pruning after training methods could merely be the fact that the former is not trained before it gets pruned as opposed to the latter yielding the observed performance gap quite obviously._
>
> We interpret this comment as posing the following hypothesis: _pruning later in training leads to higher accuracy in Section 6 because the network already has high accuracy when pruning occurs._ As a counterexample, consider random pruning (the orange line in Figure 6). If this hypothesis were true, then the performance of random pruning should also improve when pruning later in training. As Figure 6 shows, this is clearly not the case. We have added a sentence (Section 6 Paragraph 3) to specifically address this point.

---

> ### Author Response · Authors · 2020-11-24
> **Author Response (Part 1)**
>
> We thank the reviewer for their detailed feedback. We have responded to the reviewer’s comments in-line below, and we ask the reviewer to look at our revised paper where noted.
>
> ---
>
> _This paper proposes to achieve the following goal: to understand why early pruning methods perform worse than pruning after training._
>
> To clarify, this is only one of our goals. As we state in the introduction: (Goal 1) “Our purpose is to clarify the state of the art [in pruning at initialization], shed light on the strengths and weaknesses of existing methods, understand their behavior in practice, [and] set baselines for the future.”
>
> Only after we determine that these methods do not match the accuracy of pruning after training and perform similarly to magnitude pruning at initialization do we seek to (Goal 2) understand whether “there are broader challenges particular to pruning at initialization.”
>
> This review mainly focuses on Goal 2, but we wish to emphasize to the reviewer that Goal 1 is also an important contribution.
>
> ---
>
> _The paper appears to be focused on studying and debunking the early pruning methods..._
>
> These serve the dual purpose of addressing Goal 1 and providing evidence for Goal 2. We believe that our efforts toward this goal are an important contribution to the pruning literature.
>
> ---
>
> _...rather than finding out why pruning after training is better than pruning early._
>
> _This work does not explain “why all methods fall short of magnitude pruning after training in a convincing manner.”_
>
> We don't explain why all methods fall short of magnitude pruning after training. This would have been a very challenging goal. However, we believe we take important steps toward understanding the differences in behavior between four methods for pruning at initialization and a standard method for pruning after training. This provides an important foundation for further inquiry into whether it may more generally be difficult to prune at initialization and match the accuracy of pruning after training.
> We have revised the paper to make clear that our results demonstrate a correlation between accuracy and insensitivity to ablations that lays the foundation for future causal understanding of the gap between pruning after training and a suite of methods for pruning at initialization.
>
> ---
>
> _The result of random pruning...and the failure of GraSP when inverted are quite interesting, but they remain empirical without contributing directly to fulfill the purpose._
>
> _Some of the conclusions drawn from the ablation studies are not convincing (e.g., lack of specificity or insensitivity to initialization may limit performance)._
>
> We know of no SOTA weight-pruning methods in the literature that maintain their accuracy under the shuffling or reinitialization ablations. Performing these ablations on SOTA weight-pruning methods consistently leads to lower accuracy [Han et al., 2015; Frankle & Carbin, 2019; Frankle et al., 2020]. These results raise the question of whether methods that, in effect, only specify the layerwise proportions by which to randomly prune (rather than the specific connections or weights to prune) will be restricted to a lower stratum of accuracy. Since the early pruning methods reach the same accuracy regardless of whether pruning occurs randomly per-layer, they would be restricted to this lower stratum of accuracy if this is the case. Although this is not conclusive proof that it is impossible for a pruning method to reach state-of-the-art tradeoffs based on layerwise proportions alone, there are no examples of pruning methods with this property among the dozens of published methods in the literature. Our significant empirical work has uncovered these phenomena, laying the groundwork for further empirical and theoretical studies.
>
> The result of the inversion experiment (“the failure of GraSP”) addresses Goal 1: “shed[ding] light on the strengths and weaknesses of existing methods” and “understand[ing] their behavior in practice.”
>
> With respect to the concern that our results remain empirical, we are focused on the behavior of these methods in practice, and we believe that our analysis is rigorous with respect to the four methods we examine. Our results raise broader questions that could be a valuable subject for future theoretical treatment (e.g., Is randomly pruning in a layerwise manner or pruning in a manner insensitive to reinitialization inherently restricted to lower accuracy? Is it impossible for a certain class of pruning methods to prune in a manner that is sensitive to these ablations at initialization?). However, conducting this theoretical analysis is beyond the scope of our practically oriented study.
> Finally, as we note at the end of Section 5, the practical behavior of these methods differs from that predicted by the theoretical analysis of these methods in the papers that proposed them, raising questions about whether existing theory provides a productive means for analyzing these techniques.

---

### Official Review · AnonReviewer3 · 2020-10-28
**Official blind review #3**

**Rating:** 7
**Confidence:** 3

**Review:**

##########################################################################

Summary:

Generating a pruned network falls into two broad categories: 1) spend some extra time and effort to train or fine-tune the pruned model after first training a dense version, or 2) cut out that extra time and effort by generating a sparse network "from scratch."  While approach (1) has historically given the best accuracy, recent advances (such as the lottery ticket hypothesis) suggest that there are sparse networks hidden in the initialization that don't need to first be trained, if only we could divine the structure of those models.  Approach (2) seeks to do just this: determine the connectivity as close to initialization possible.  However, even the best results taking this second path fall short when compared to the accuracy of the former path - why is this?  The submission pokes at three recent techniques to pull out some commonalities that are *not* shared with (1), suggesting possible issues that need to be overcome to improve accuracy, and proposes a set of experiments and comparisons that should be part of any new technique that claims to discover a good sparse mask at initialization.

##########################################################################

Reasons for score:

Overall, this submission raises some important questions (why is from-scratch pruning falling short of SOTA accuracy?), empirically shows the performance of some leading techniques (and that they fall short in a well-motivated range of sparsities), and, via ablation studies, points towards some potential reasons.  More importantly, these findings are interesting:
- There's no single SOTA method for sparse-from-scratch training
- There's a need for consistent reporting in this area, and the ablation studies performed have been shown to lead to useful insights; adopting them as standard for future work seems fruitful.

I think these are enough to warrant an "above-the-threshold" rating, but a higher rating would require empirical results on more networks on large-scale data sets, or the inclusion of techniques the submission itself suggests might be a promising direction forward in the current gauntlet of tests.

##########################################################################

Pros:

+ The organization of the paper makes it easy to follow the logical progression and points being raised.
+ The direct comparisons of three recent techniques (SNIP, GraSP, and SynFlow) on different networks and data sets fills in some gaps in the literature.
+ Further, the ablation studies performed on these techniques yield surprising results, both in isolation (inverting GraSP improves accuracy!) and when compared to magnitude pruning after training (these three are invariant to shuffling and re-initialization).

##########################################################################

Cons:

- Experiments on large data sets are limited to RN50 on ImageNet.
- The three particular techniques chosen (SNIP, GraSP, SynFlow) aren't particularly motivated - why these three, and not other recent techniques?  (A potential answer may be that there's no training before pruning is finished, but why is this important?)
- Mostly tongue-in-cheek: the submission doesn't answer all the questions it raises, unfortunately.

##########################################################################

Questions:

- What benefit do the three at-initialization "static" pruning techniques have over those that reduce training FLOPs and memory requirements but allow the sparse mask to change dynamically, like RigL (Evci et al., 2020) and sparse momentum (Dettmers and Zettlemoyer, 2019)?  Is there a reason they do not belong in the current lineup?  The overhead of occasionally updating the mask shouldn't be too imposing.

- In the final paragraph, it is suggested that it may be tricky to compare the training cost of "a method that prunes to sparsity s at step t against a method that prunes to sparsity s' < s at step t' > t.  If method A prunes to a higher sparsity at an earlier time step, shouldn't it cost strictly less than method B, which prunes to a lower sparsity later in training?

##########################################################################

Minor suggestions:

Figure 2 is never referenced in the text (that I could find), and LTR isn't defined until the following page.


##########################################################################

Updates are appreciated

Hi, Authors,

I appreciate the updates you've made to the paper and the responses to my questions.  You're quite correct that RN50 and ImageNet is sufficient to illustrate deficiencies.

(I'd still want to see broader experiments for claims of some new method overcoming these deficiencies, though!  When broadening scope to other tasks, I'd expect the authors of prior methods designed for vision tasks would be okay with use of their methods as baselines if there are no methods designed specifically for those new tasks.)

With this in mind, I'll update my rating to a 7.

---

> ### Author Response · Authors · 2020-11-19
> **Author Response**
>
> We thank the reviewer for the detailed feedback. We have responded to the reviewer’s comments in-line below.
>
> ---
>
> _A higher rating would require empirical results on more networks on large-scale datasets_
> _Experiments on large data sets are limited to RN50 on ImageNet._
>
> When it comes to networks, datasets, and scale, our paper is the most comprehensive study ever conducted on the topic of pruning neural networks at initialization, and we pushed our resources and our budget to the limit in order to make this happen.
>
> In the main body of our paper alone, we present results from training more than 11,000 networks. This includes training about 600 networks (14 sparsities) on ImageNet; for comparison, the GraSP paper (Wang et al.) is the only paper on pruning at initialization that includes any results on ImageNet, and it contains just 36 networks (3 sparsities) trained on the task.
>
> Since our purpose was to rigorously evaluate and examine these techniques, we chose to focus our limited resources on fewer settings in depth rather than more settings in a shallower manner. The papers introducing the methods we look at primarily (SNIP) or only (GraSP, SynFlow) studied image classification; our work therefore assesses these methods in their original domains. We were cognizant that reviewers or authors might consider it unfair if we included other computer vision tasks or natural language processing tasks for which these techniques were not designed.
>
> Although we agree that any empirical paper is stronger if it includes more settings, we believe the current experiments are convincing and comprehensive and we lack the resources to add more large-scale settings.
>
> ---
>
> _Inclusion of techniques the submission itself suggests might be a promising direction forward in the current gauntlet of tests._
>
> We intentionally restricted our paper to analysis of existing techniques. We wanted to avoid confusing the purpose of our paper by mixing analysis of existing techniques with an entirely new proposal. Even by restricting our scope in this way, the present results still fill 8 pages in the main body and 27 pages of appendices.
>
> ---
>
> _The three particular techniques chosen (SNIP, GraSP, SynFlow) aren’t particularly motivated. Why these three, and not other recent techniques?_
>
> To the best of our knowledge, SNIP, GraSP, SynFlow, and NTT are the only published methods specifically designed to prune neural networks at initialization. We did not include NTT because the original paper only evaluated the technique on small networks for MNIST and CIFAR-10 (smaller than the smallest setting we consider).
>
> As we discuss in the related work (Section 2), the only other proposed methods for pruning at initialization that we could find at the time of submission are iterative variations of SNIP (de Jorge et al. 2020 and Verdenius et al. 2020). We consider iterative SNIP in Appendix G.
>
> ---
>
> _Is there a reason [dynamic sparsity techniques] do not belong in the current lineup?_
>
> Although dynamic pruning methods (e.g., Mocanu et al., Dettmers & Zettlemoyer, Evci et al.) are an exciting direction for training sparse networks from the start, they are beyond the scope of this work. Our goal was to study the growing literature on pruning at initialization and to highlight the apparent difficulty of doing so; dynamic pruning methods begin with a sparse network, but they mainly prune after initialization, so they do not fall into this category. Many of our experiments do not make sense in the context of dynamic pruning, for example reinitializing after pruning (Section 5) and pruning only at a specific point in training (Section 6).
>
> ---
>
> _The final paragraph._
>
> This was a typo. One of the < signs was backwards. We have corrected this mistake in the updated manuscript (to be posted shortly).
>
> ---
>
> _Figure 2 is never referenced in the text._
>
> This was an editing mistake, and we appreciate that you brought this to our attention. We are re-inserting a reference to this figure in the introduction (updated manuscript to come shortly). We intend Figure 2 to serve as a quick reference for the experiments we perform and how they differ from the experiments in prior work.

---

> > ### Author Response · Authors · 2020-11-24
> > **Paper is Updated**
> >
> > We wanted to notify you that we have posted our revised version of the paper. As promised, we have addressed the typo in the final paragraph and have reintroduced a reference to Figure 2 in the introduction. We have also made many changes throughout to address comments from other reviewers.

---

### Official Review · AnonReviewer4 · 2020-10-30
**Some conclusions of this paper are unsubstantiated or should be rephrased**

**Rating:** 6
**Confidence:** 5

**Review:**

## Summary

The paper provides an extensive empirical analysis of Pruning-at-Initialization (PaI) techniques and compares it against two pruning methods after (or during) training. This comparison sheds some light on why pruning at initialization is inherently hard. Furthermore, the comparison among PaI methods with various ablations shows some inherent properties that are common to PaI methods and the benefits/drawbacks of certain methods. With these experiments, certain conclusions are reached among them an important one is that PaI methods only determine what is the fraction of weights to be pruned in each layer rather than which weights to prune.

## Strengths

1. Quantitative comparison of PaI methods is an important contribution to the community. In addition, the paper extensively analyses them with various ablations (eg, shuffle the layerwise masks, reinitialization) which uncovers various properties of PaI methods previously not conveyed clearly in the respective papers.

2. Extensive analysis shows that there is a performance gap of PaI methods compared to Pruning-after-Training (PaT) methods (understandably) and attempts to explore the potential reasons. This analysis uncovers some inherent difficulties of PaI. The analysis provided in this paper could serve as a guide to better understand and improve PaI methods.

3. Overall the paper is clearly written and sufficient details are provided in the appendix.

## Weaknesses

The main weaknesses of the manuscript in my opinion are as follows:

1. Some conclusions of this paper are unsubstantiated or should be rephrased:
	- First, the experiments mainly convey the inherent difficulties of PaI rather than the issues with specific methods themselves. To address this concern, the methods (SNIP, GraSP, SynFlow) are performed during training and showed that they perform inferior to LTR in Fig. 6. However, this comparison is unfair as LTR has additional information which is obtained after training (pruning mask is obtained after training but applied during training) and those PaI methods are not specifically designed to be performed during/after training (even so some perform reasonably well). In short, it is not clear that the performance of PaT can be matched by PaI given only the information at initialization. Note that, having access to the pruning mask after training defeats the purpose of PaI since given an already trained network, one might as well simply prune at the end (no need to retrain from scratch). I believe, Sec. 7 has some discussions about this (not complete) but should come early in the paper and emphasized. Please clarify and rephrase certain parts (especially in the introduction).
	- One of the main conclusions: "PaI methods only determine what is the fraction of weights to be pruned in each layer rather than which weights to prune" should be rephrased. In fact, it is possible to have a pathological case (eg, disconnected layers at a given pruning ratio) where knowing the optimal pruning ratio is not sufficient to obtain matching accuracy. These pathological cases are not observed in practice due to the random component (initialization or shuffling) and it should be emphasized. Also, since the weight initialization is iid Gaussian, there may not be sufficient information for PaI methods to select each weight individually but rather select them as a group in each layer or whole network.
	- Another hypothesis that the performance gap between PaI and PaT is due to the robustness of PaI methods to shuffling, reinitialization, etc, are also not clear. I understand that there is a correlation exists that some PaT methods are not robust to such variations. But it is NOT clearly demonstrated that being robust to such variations is necessarily a bad thing for PaI. In fact, one would think it is a good thing that PaI methods are robust to such variations given that there is not a lot of information available at initialization (note initialization is iid) to perform effective pruning and it seems these methods are robust and perform competitively to unpruned networks.

2. Inconsistency of SNIP being independent to initialization:
	- In the last paragraph of page 5, it is mentioned that SNIP is independent to what initialization is used. However, there is a follow-up work of SNIP showing that it could be beneficial for SNIP if the network is initialized to have good signal propagation [a]. Please discuss this in the context to avoid any misinterpretations.

3. Issue with random shuffling:
	- The random shuffling experiment needs some refinement in my opinion. I believe PaI methods such as SNIP, GraSP or SynFlow might have some unpruned weights which are not updated during training (ie, they are disconnected in the signal propagation path). In my personal experience, I observed that there are about 1-2% of unpruned disconnected weights in the network (in particular layers this value could be up to 10%) for SNIP (meaning they can be removed) depending on the network. This means the effective sparsity is slightly lower than what is observed in SNIP (not mentioned in the original paper though) and presumably in other methods as well. This observation would be applicable to random shuffling as well. I mean, after random shuffling there might be some unpruned disconnected weights and they could be removed before training, leading to higher effective sparsity. Furthermore, the existence of unpruned disconnected weights could be the reason for similar behaviour even after random shuffling in each layer.

I found the experiments in this paper to be thorough but I think the deduced conclusions are slightly off with respect to the observations in the experiments. I believe this paper will be a good contribution to the pruning literature if the conclusions are tightened.

## Minor Comments

1. Related work: SNIP is not a follow-up of LTH but rather they are concurrent works published in ICLR 2019.
2. Abstract: I think "undermines" might be a strong word given that there are questions regarding deduced conclusions.

## References

- [a] Lee, N., Ajanthan, T., Gould, S. and Torr, P.H., 2020. A signal propagation perspective for pruning neural networks at initialization. ICLR.

---

> ### Author Response · Authors · 2020-11-24
> **Author Response (Part 3)**
>
> _The random shuffling experiment needs some refinement in my opinion. I believe PaI methods such as SNIP, GraSP, or SynFlow might have some unpruned weights which are not updated during training._
>
> We interpret this comment to mean the following:
>
> > _It is possible that SNIP, GraSP, and SynFlow leave some weights unpruned but disconnected, which means the “effective sparsity” (the sparsity when also eliminating disconnected weights) of these networks is higher than the “actual sparsity” (the sparsity when taking into account connections that are explicitly pruned). Furthermore, when randomly shuffling, some of these disconnected weights may become reconnected. As such, random shuffling might appear to maintain performance simply because it has a lower effective sparsity than the unmodified pruning technique._
>
> To evaluate this hypothesis, we added Appendix N, in which we compare the effective sparsities for SNIP, GraSP, and SynFlow with and without randomly shuffling.
>
> **TLDR: At the highest matching actual sparsity, the effective sparsity of the shuffled network was larger by a negligible amount: at most 1.0026x (i.e., 0.26%) larger than the unmodified network and in most cases closer to 1.0005x (i.e., 0.05%) larger. At more extreme sparsities, this ratio was higher but still small: at most 1.0353x (i.e., 3.5% larger) and in most cases closer to 1.003x (i.e., 0.3%) larger.**
>
> **Full details:**
> To determine which weights were disconnected, we:
> 1. Set each weight in the network to 1 (if unpruned) or 0 (if pruned).
> 2. Forward-propagated a single example comprising all 1’s.
> 3. Computed the sum of the logits.
> 4. Computed the gradients with respect to this sum.
> 5. Prune any unpruned weight with a gradient of 0.
>
> For each “actual sparsity,” we computed the ratio of $\frac{\text{effective parameters in the shuffled network}}{\text{effective parameters in the unmodified network}}$.
>
> A summary of our results is below (see Appendix N for full data). Please refer to the TLDR above for our summary of the data. We conclude that, although random shuffling does often restore some disconnected parameters, the actual fraction of parameters it restores is minuscule at and beyond the matching sparsities we focus on. These differences are so small that the effective sparsities round to the same values as the actual sparsities on the x-axis labels of our plots.
>
> | Network                  | Pruning Method | Ratio at Highest Matching Sparsity | Ratio at Highest Sparsity We Include |
> |--------------------------|----------------|------------------------------------|--------------------------------------|
> | ResNet-20 (CIFAR-10)     | Magnitude      | 1.0000                             | 1.0016                               |
> |                          | SNIP           | 1.0006                             | 1.0353                               |
> |                          | GraSP          | 1.0000                             | 1.0197                               |
> |                          | SynFlow        | 1.0000                             | 0.9921                               |
> | VGG-16 (CIFAR-10)        | Magnitude      | 1.0000                             | 0.9998                               |
> |                          | SNIP           | 1.0007                             | 1.0058                               |
> |                          | GraSP          | 1.0003                             | 1.0036                               |
> |                          | SynFlow        | 1.0000                             | 1.0001                               |
> | ResNet-18 (TinyImageNet) | Magnitude      | 1.0003                             | 1.0000                               |
> |                          | SNIP           | 1.0026                             | 1.0227                               |
> |                          | GraSP          | 1.0011                             | 1.0030                               |
> |                          | SynFlow        | 1.0001                             | 1.0001                               |
> | ResNet-50 (ImageNet)     | Magnitude      | 1.0000                             | 0.9979                               |
> |                          | SNIP           | 1.0003                             | 1.0016                               |
> |                          | GraSP          | 1.0000                             | 1.0009                               |
> |                          | SynFlow        | 1.0000                             | 1.0001                               |

---

> ### Author Response · Authors · 2020-11-24
> **Author Response (Part 2)**
>
> _Since the weight initialization is iid Gaussian, there may not be sufficient information for pruning methods to select each weight individually but rather to select them as a group in each layer or the whole network._
>
> We are uncertain about the specific technical weakness in our paper that this comment refers to and we would appreciate clarification.
>
> We agree that this is a possible hypothesis for why it may be difficult or impossible to prune in a weight-specific or initialization-specific manner at initialization. However, we do not understand why this hypothesis is listed as a weakness of our paper: it would answer our question about the possible difficulty of pruning in a weight-specific or initialization-specific manner at initialization (Section 1 Paragraph 11, Section 7 Paragraph 7), and we do not make any claims about the specific mechanism that makes it difficult to do so at initialization (a point we specifically discuss in the newly-added Section 7 Paragraph 8).
>
> ---
>
> _Another hypothesis that the performance gap between PaI and PaT is due to the robustness of PaI methods to shuffling, reinitialization, etc, are also not clear. I understand that a correlation exists that some PaT methods are not robust to such variations. But it is NOT clearly demonstrated that being robust to such variations is necessarily a bad thing for PaI_
>
> We have revised the paper to make clear that our results demonstrate a correlation between accuracy and insensitivity to ablations that lays the foundation for future causal understanding of the gap between pruning after training and these methods for pruning at initialization.
>
> We know of no SOTA methods for pruning weights after training in the literature that can maintain its accuracy under these ablations. Performing these ablations on these methods consistently leads to lower accuracy [Han et al., 2015; Frankle & Carbin, 2019; Frankle et al., 2020]. These results certainly raise the question of whether methods that, in effect, only specify the layerwise proportions in which to randomly prune (rather than the specific connections or weights to prune) will be restricted to this lower stratum of accuracy (which we interpret as a bad thing for pruning at initialization). More broadly, this raises the question of whether it may be impossible to prune at initialization in a way that could not be replaced by randomly pruning at the same per-layer rates, in which case, on networks like ResNet-50 on ImageNet, there may be inherent trade-offs to pruning at initialization (since existing methods cannot match full accuracy at any appreciable sparsity).
>
> ---
>
> _In fact, one would think it is a good thing that PaI methods are robust to such variations given that there is not a lot of information available at initialization (note initialization is iid) to perform effective pruning and it seems these methods are robust and perform competitively to unpruned networks._
>
> It is possible that maintaining accuracy under the ablations may be a benefit in some way, but this is orthogonal to the implication that methods that maintain accuracy under these ablations may be restricted to a lower stratum of accuracy.
>
> As to whether these methods perform competitively to unpruned networks, it depends on the definition of “competitively.” Due to the subjectivity of this term, we focus on whether methods can match the accuracy of the unpruned networks. Our experiments in Section 4 show that it is not the case that these methods perform “competitively” according to this criterion: for example, on the most realistic benchmark (ResNet-50 on ImageNet), none of these methods can reach full accuracy at any sparsity we consider.
>
> ---
>
> _In the last paragraph of page 5, it is mentioned that SNIP is independent to what initialization is used. However, there is a follow-up work of SNIP showing that it could be beneficial for SNIP if the network is initialized to have good signal propagation [a]._
>
> Thank you for bringing this work to our attention. We have updated this paragraph to clarify that we are only describing one specific initialization in our experiment (“even when the initialization is not informative” -> “in a case where the initialization is not informative”), and we have added a footnote noting the relationship between SNIP and initialization as described in the reference you provide.
>
> ---
>
> _SNIP is not a follow-up of LTH._
>
> We have updated the text accordingly - thank you for this clarification.
>
> ---
>
> _I think “undermines” might be a strong word._
>
> We have removed all language of this kind from our paper, particularly in the abstract and the last paragraph in Section 5. We would appreciate any follow-up suggestions you have for further improving the way we address this point.

---

> ### Author Response · Authors · 2020-11-24
> **Author Response (Part 1)**
>
> We thank the reviewer for their knowledgeable comments and detailed suggestions. We have taken these comments very seriously, and we have revised the paper accordingly. We would appreciate it if the reviewer would look at our responses below and our updated paper in detail.
>
> ---
>
> _The experiments mainly convey the inherent difficulties of PaI rather than the issues with the specific methods themselves._
>
> We agree that our results do raise the question of whether there are inherent difficulties to pruning at initialization and that this question is an important takeaway of our paper.
>
> At the same time, our experiments are only able to discern challenges specific to the methods that we study, and we cannot make strong claims about inherent difficulties of pruning at initialization based on these results. The difficulties may be due to the methods themselves rather than to inherent difficulties of pruning at initialization.
>
> We have revised the paper to make this narrative clear (Section 1 Paragraph 11, Section 7 Paragraph 6).
>
>
> ---
>
> _This comparison [in Section 6]  is unfair as LTR has additional information which is obtained after training._
>
> _PaI methods are not specifically designed to be performed during/after training._
>
> We have made changes throughout the paper to address these concerns. In particular:
>
> 1. We have explicitly clarified that SNIP, GraSP, and SynFlow were not designed to prune after initialization (Section 1 Paragraph 13, Section 6 Paragraph 2, Section 7 Paragraph 12) and that we are evaluating them outside of the setting for which they were designed (Section 6 Paragraph 2).
> 2. We have clarified that, for the experiment in Section 6, we are interested in whether accuracy improves and whether the methods become sensitive to the ablations, not the specific level of final accuracy (Section 6 Paragraph 2).
> 3. We have removed any judgment about which methods are SOTA after initialization, since the methods were not designed for this purpose (deleted from Section 7). We have also removed any statements that imply performance that is lower than LTR reflects a weakness of the methods (deleted from Section 1 and Section 6).
> 4. The remaining comparisons to the performance of LTR (Section 6 Paragraph 5, Section 7 Paragraph 12) are stated with the context that it may be impossible to reach this performance in practice since LTR has access to information from the end of training.
> ---
>
> _It is not clear that the performance of PaT can be matched by PaI given only the information at initialization._
>
> _I believe Sec. 7 has some discussions about this but should come early in the paper and emphasized._
>
> We have made the following changes to address this concern:
> 1. We have removed all statements in the paper that set the expectation that the techniques for pruning at initialization should be able to match the performance of pruning after training (deleted from the following locations in the original manuscript: Section 1 Paragraph 6, Section 1 Paragraph 11, Section 3 Paragraph 9, Section 7 Paragraph 6).
>
> 2. We have also removed all statements that imply that it is a shortcoming of the methods that they do not match this performance (deleted from the following locations in the original manuscript: Section 1 Paragraphs 7,8,9,11, Section 4 Paragraph 5, Section 5 Paragraph 1); in particular, the phrases “fall short” and “underperform” no longer appear in the paper
>
> 3. We have revised Section 1 to make clear that it may not be possible to match the performance of pruning after training by pruning at initialization (in the updated manuscript: Section 1 Paragraph 9).
>
> 4. In place of these comparisons, we have added a new research question: “are there broader challenges particular to pruning at initialization?” (Section 1 Paragraph 7).  This question was already implicit in the paper, but it was previously conflated with the question of whether methods for pruning at initialization could match methods for pruning after training. Thanks to your feedback, we have teased apart these questions, which we believe significantly improves the clarity of the manuscript.
> ---
> _One of the main conclusions: "PaI methods only determine what is the fraction of weights to be pruned in each layer rather than which weights to prune" should be rephrased. It is possible to have a pathological case (e.g., disconnected layers at a given pruning ratio) where knowing the optimal pruning ratio is not sufficient to obtain matching accuracy.  These pathological cases are not observed in practice due to the random component (initialization or shuffling) and it should be emphasized._
>
> We have added a footnote in Section 5 Paragraph 3 (where we first run the random shuffling experiment) to clarify that, at extreme sparsities or under an unlucky random draw, it is possible that shuffling could lead to lower accuracy (although we do not ever observe this behavior ourselves for the settings and sparsities we consider).

---

### Decision · Program_Chairs · 2021-01-07
**Final Decision**

**Decision:**

Accept (Poster)

**Comment:**

The paper analyses several approaches to pruning at initialization, compared to after training. There was a large gap in reviewers appreciation of the paper, but I think that the pros outdo the cons as the paper show a lot of insights overall. I recommend accepting the paper.